# Drainage explains soil liquefaction beyond the earthquake near-field

Shahar Ben-Zeev [1,2] ✉, Liran Goren [3], Renaud Toussaint [2,4] & Einat Aharonov [1,5]

Earthquake-induced soil-liquefaction is a devastating phenomenon associated with loss of soil rigidity due to seismic shaking, resulting in catastrophic liquid-like soil deformation. Traditionally, liquefaction is viewed as an effectively undrained process. However, since undrained liquefaction only initiates under high energy density, most earthquake liquefaction events remain unexplained, since they initiate far from the earthquake epicenter, under low energy density. Here we show that liquefaction can occur under drained conditions at remarkably low seismic-energy density, offering a general explanation for earthquake far-field liquefaction. Drained conditions promote interstitial fluid flow across the soil during earthquakes, leading to excess pore pressure gradients and loss of soil strength. Drained liquefaction is triggered rapidly and controlled by a propagating compaction front, whose velocity depends on the seismic-energy injection rate. Our findings highlight the importance of considering soil liquefaction under a spectrum of drainage conditions, with critical implications for liquefaction potential assessments and hazards.

Seismically induced soil liquefaction is a natural hazard that commonly occurs during earthquakes[1]. During liquefaction, a soil that initially possessed an elasto-plastic rheology and was capable of supporting the load of infrastructure, loses its strength and stiffness in response to earthquake shaking, consequently exhibiting fluid-like rheology. Earthquake-induced soil liquefaction results in buildings and infrastructures sinking[2], floating and tilting[2], ground lateral spreading[2], settlement[3], and landsliding[4]. Liquefaction damage often leads to extensive human casualties[4,5], destruction of lifelines[2,6], and economic losses[6–9], that may result in complete abandonment of formerly inhabited areas[7], posing a significant challenge to community resilience[10].

The classical mechanism explaining seismically induced soil liquefaction[1] considers the soil as an effectively undrained medium. Upon cyclic shear, an initially loosely-packed soil tends to reduce its pore volume, as readily documented under dry and drained conditions[11]. If the pore fluid flow rate is slow compared to the rate of porosity reduction, as is expected in an undrained soil response, the pore fluid is trapped within the contracting pores and its pressure increases. If the pore pressure builds up to the level of the overburden stress (commonly lithostatic values), then the effective stress reduces to zero[12], the soil loses its shear strength and stiffness and is said to be liquefied[8,11]. Undrained lab experiments[8,11,13,14] showed that during continuous shaking, and depending on the initial soil density and the applied shear stress, the pore pressure builds up gradually and reaches lithostatic values after several to tens of shear cycles.

Despite the overall success of the undrained perspective in describing the conditions leading to pore pressure rise and soil strength and stiffness loss during earthquakes, it struggles to explain soil liquefaction beyond the near-field, far from the earthquake's epicenter, where the seismic energy density input is small. Empirical inferences established a lower bound of 30 J m$^{-3}$ for the seismic energy density required to induce liquefaction by undrained consolidation[15,16]. Consequently, as the seismic energy decays away from the

[1]Institute of Earth Sciences, The Hebrew University of Jerusalem, 91904 Jerusalem, Israel. [2]University of Strasbourg, CNRS, ENGEES, Institut Terre & Environnement de Strasbourg, UMR7063, F-67000 Strasbourg, France. [3]The Department of Earth and Environmental Sciences, Ben-Gurion University of the Negev, 84105 Negev, Israel. [4]PoreLab, the Njord Centre, Department of Physics, University of Oslo, P.O. Box 1048 Blindern, NO-0316 Oslo, Norway. [5]Departments of Geosciences and Physics, The Njord Centre, University of Oslo, Oslo, Norway. ✉e-mail: shahar.benzeev@mail.huji.ac.il

earthquake's epicenter, liquefaction events beyond the near-field should become uncommon. Nonetheless, the majority of the events in an extensive soil liquefaction compilation[16,17] were triggered beyond the earthquake near field, at a distance greater than one fault rupture length from the hypocenter, where the seismic energy density is well below the 30 J m$^{-3}$ threshold and as low as 0.1 J m$^{-3}$ (Fig. 1 in ref. [16]). The discrepancy between the leading theory and field observations of soil liquefaction indicates that our understanding of the conditions and processes associated with earthquake-induced soil liquefaction is incomplete.

The main attempt to reconcile theory and observations[16] invoked seismically induced permeability enhancement[7,18] between deep pressurized aquifers and the liquefied layer. However, dynamic permeability increase and the availability of buried high fluid-pressure sources represent unique geometric and hydrologic conditions, likely precluding it from being a general mechanism for liquefaction beyond the near field.

In a rare movie capturing soil liquefaction[19] at the Makuhari Seaside Park in Chiba (Japan), during the Tohoku earthquake (2011)[20], the photographer commented while pointing to the lawn: "... there was water just coming up right there, on the ground... and the ground is just swaying right now". This testimony suggests that fluid drainage toward the surface during an earthquake could play an important role in the process of soil liquefaction, which agrees with recent theoretical and experimental studies proposing that rapid fluid flow could be instrumental in initiating liquefaction[21–28]. This implies that the undrained liquefaction initiation mechanism does not necessarily cover the full spectrum of conditions leading to soil liquefaction.

Drained liquefaction initiation implies that the timescale of fluid flow is shorter than the timescale associated with earthquake-induced soil deformation. In this scenario, porous fluid flow toward a drained boundary is accompanied by pore pressure gradients that exert seepage forces on the soil grains, supporting their weight, weakening grain contacts, and reducing soil strength. The notion that pressure gradients and seepage forces could fully support grains is, in itself, not novel. Static vertical pressure gradients supporting a layer of grains is known as quicksand conditions[29,30]. Similarly, interstitial fluid ejection leading to ground settlement is a known post-liquefaction failure mechanism[31–33].

The evolution of the pore pressure in a deformable saturated granular media can be described by a diffusion equation with a source term related to the granular skeleton deformation[21,22,27,34]:

$$\frac{\partial P'}{\partial t} - \frac{1}{\beta_f \eta \phi} \mathbf{\nabla} \cdot [\kappa \mathbf{\nabla} P'] + \frac{1}{\beta_f \phi} \mathbf{\nabla} \cdot \mathbf{u_s} = 0, \qquad (1)$$

where $P'$ is the dynamic pore pressure deviation from hydrostatic value ($P' = P - P_{hyd}$), $\beta_f$ and $\eta$ are the fluid compressibilty and viscosity, respectively, $\kappa$ is the permeability, $t$ is time and $\mathbf{\nabla}$ is a spatial derivative. The second term in Eq. (1) is a diffusion term arising from Darcy flow, while the third term describes the internal source for dynamic pore pressure, due to divergence of solid grain velocities ($\mathbf{u_s}$). This term can be approximated[22,27] as the rate of pore space compaction and dilation, $\mathbf{\nabla} \cdot \mathbf{u_s} \simeq \frac{1}{1-\phi} \frac{\partial \phi}{\partial t}$, where $\phi$ is the porosity.

Since fluid drainage within the soil granular media is expected to obey Darcy's flux law, the characteristic velocity scale in Eq. (1) is identified with $u_0 = (\kappa_0/\phi\eta) \cdot (\sigma_0^h/h)$, where $\sigma_0^h/h$ describes the initial effective lithostatic stress gradient, which is also the pressure gradient during liquefaction. $\sigma_0^h$ is the initial effective lithostatic stress at depth $h$, and $\kappa_0$ is the characteristic permeability. There are two length scales characterizing the system: Stress and pressure change gradually over the layer depth, $h$, yet grain divergence and convergence could occur over a different length scale, $l$, which could be as small as several grains wide[21,27]. Subsequently, non-dimensional parameters (represented by caret symbols, ˆ) can be defined as follows: $\mathbf{\nabla} = \hat{\mathbf{\nabla}}_l/l$ where $\mathbf{\nabla}$ appears as

a divergence operator, $\mathbf{\nabla} = \hat{\mathbf{\nabla}}_h/h$ where $\mathbf{\nabla}$ appears as a gradient operator, $\hat{\mathbf{u}}_s = \mathbf{u}_s/u_0$, $\hat{t} = t/t_0$, $\hat{P} = P'/\sigma_0^h$, and $\hat{\kappa} = \kappa/\kappa_0$. Eq. (1) can then be re-written as:

$$\text{De} \frac{\partial \hat{P}}{\partial \hat{t}} - \hat{\mathbf{\nabla}}_l \cdot [\hat{\kappa} \hat{\mathbf{\nabla}}_h \hat{P}] + \frac{1}{\phi} \hat{\mathbf{\nabla}}_l \cdot \hat{\mathbf{u}}_s = 0 \qquad (2)$$

The non-dimensional coefficient in front of the first term in Eq. (2) is known as the Deborah number (De)[21,22,27,35,36]:

$$\text{De} = \frac{t_d}{t_0} = \frac{hl\beta_f\eta\phi}{T\kappa_0}, \qquad (3)$$

which expresses the ratio between the timescale for pressure diffusion $t_d = \frac{hl}{D}$, where $D = \frac{\kappa_0}{\beta_f\eta\phi}$ is the pore pressure diffusion coefficient, and the timescale of deformation imposed by the shaking period, $t_0 = T$. Therefore, the De number provides a metric for evaluating the system's drainage conditions. When De $\ll 1$ and $t_d \ll T$, pore pressure diffusion is sufficiently rapid, so as to allow drainage during shaking. In this case, which we term "drained", the first term in Eq. (2) becomes negligible, and the diffusion (second term) balances the source term (third term). When De $\gg 1$, the layer is "undrained", the diffusion term is negligible, and the source term is balanced by the temporal derivative of the dynamic pressure[21]. Notably, contrary to previous studies that used the term 'drained' to describe an end-member with no change in fluid pressure[8,37], in the current study, 'drained' implies De $\ll 1$, and pore pressure gradients could emerge in response to skeleton deformation. For a discussion on drainage-related terminology and its relation to Eq. (2), see Supplementary Note 1.

To apply this general formulation to the deformation of a shallow soil column, we consider a layer of saturated cohesionless grains with a free and drained surface (where the pressure is maintained at a constant value, $P = 0$) and where no internal permeability barriers are present (Fig. 1). When such a soil layer is relatively loosely packed, with initial porosity $\phi_0$, and when it is subjected to horizontal shaking, an upward propagating compaction front (also referred to as a "solidification front") develops[27,31,38,39]. The propagating compaction front separates two regions within the layer[27] (Fig. 1a, b): a lower region, in which grains have compacted to porosity $\phi_c < \phi_0$ and are approximately stationary in the vertical direction. In this lower region, the pore pressure gradient is nearly hydrostatic, although the pore pressure itself is elevated to the value of the pressure at the front (Fig. 1c). In the region above the front, grains continuously settle at a uniform velocity while maintaining their initial porosity, $\phi_0$. The settling grains exchange place with the upward flowing pore fluid. The pore fluid pressure gradient above the front can, and normally will, become as high as lithostatic (Fig. 1c and Supplementary Note 2). This fluid pressure gradient is the source of upward directed seepage forces that support the settling grains. Under these conditions, the upward front velocity, $u_{\text{front}}$, is theoretically predicted to be[27]:

$$u_{\text{front}} = \frac{\phi_0 - 1}{\phi_0 - \phi_c} \frac{\kappa_0}{\eta} \frac{d\sigma_0}{dz}, \qquad (4)$$

where $d\sigma_0/dz$ is the effective static normal stress gradient (Fig. 1).

The uniform porosities above ($\phi_0$) and below ($\phi_c$) the front imply that active compaction occurs only across the relatively narrow front, which acts as an in-situ, migrating, pressure source, continuously forcing upward fluid drainage[27] (Fig. 1).

Here we show that the drained liquefaction mechanism, with its associated upward migrating compaction front, predicts liquefaction events triggered under low seismic-energy density. Thereby, the drained mechanism provides a general model for explaining the vast number of liquefaction events triggered beyond the earthquake near-field. This, in turn, has critical implications for the physics of

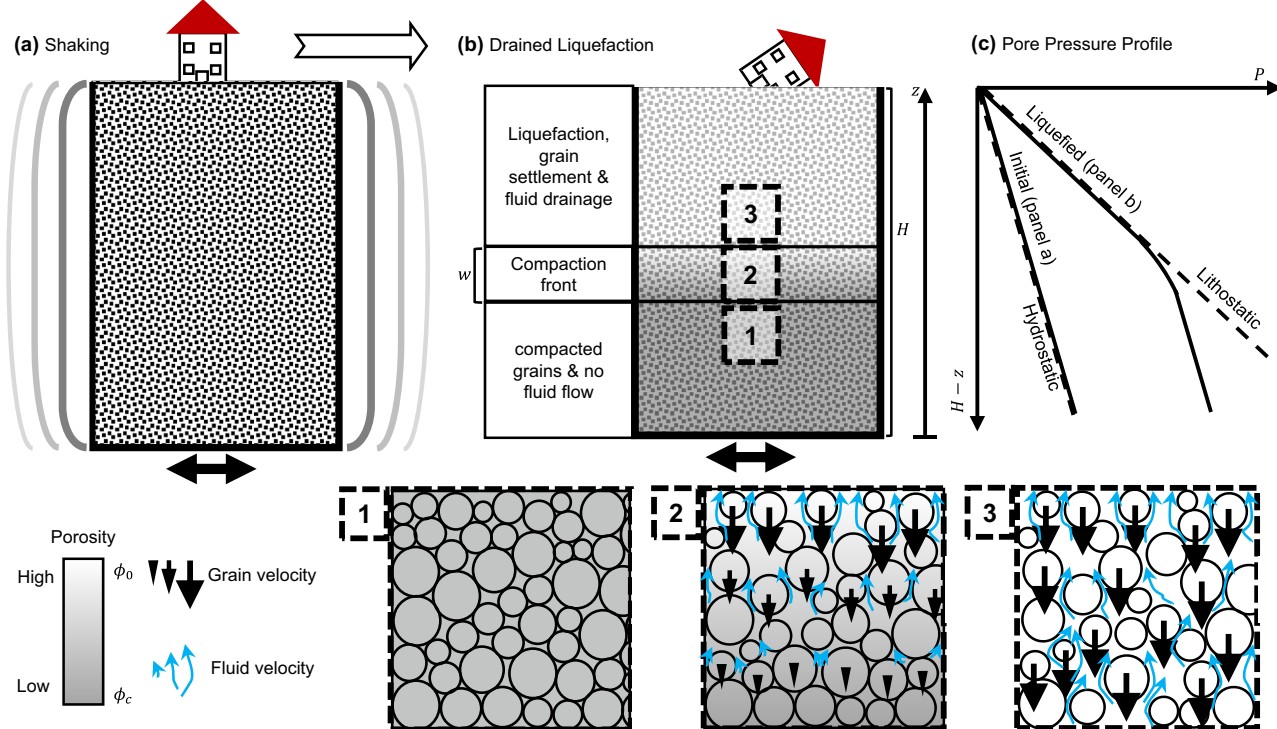

**Fig. 1 | Schematics of the compaction front dynamics that develop under drained conditions. a** An earthquake shakes a saturated soil layer of initial thickness $H$, and porosity $\phi_0$. **b** A compaction front swipes upwards, separating a [1] compacted region with porosity $\phi_c$, and negligible grain vertical velocity and fluid flow, from [3] a liquefied region with porosity $\phi_0$, an upwards fluid flow and a downwards grain settling velocity. The transition between the regions occurs gradually along a compaction front [2], whose width is $w$. **c** The pore pressure gradient is initially hydrostatic. After the formation of the compaction front and the initiation of liquefaction, the pore pressure gradient above the front is equal to the lithostatic stress gradient. Below the front, the pressure gradient is hydrostatic (which causes no fluid flow), although the pressure itself is elevated.

liquefaction, the conditions for liquefaction triggering, and consequently, the evaluation of liquefaction potential and associated hazards.

## Results and discussion

We performed grain-scale simulations and experiments of horizontally shaken layers of water-saturated cohesionless grains with a free surface. The simulations used a coupled Discrete Element Method (DEM) - Computational Fluid Dynamics (CFD) model[40], and the experiments were conducted in a transparent box, allowing inferences of grain motion and measurements of pore pressure by using an array of pressure transducers (see Methods). The simulations and experiments show the dynamics predicted by the compaction front model.

### Evaluating the drainage conditions

The drainage conditions in the simulations and experiments were evaluated by estimating the De number following equation (3). The length-scale controlling pressure gradients (and thus fluid fluxes) is conservatively chosen here as $h = H$, where $H$ is the layer height[41]. In situations of homogeneous compaction and dilation[41], $H$ would also control the divergence of grain motion. However, when a compaction front is present, grain compaction and dilation are localized[36] at the front. The natural length scale that thus emerges for the divergence of grain motion is $l = w < H$, where $w$ (Fig. 1) is the width of the compaction front. The simulations and experiments show that $w$ spans several tens of grain diameters ($\approx 20$). As a consequence, the maximal value of De is $\sim 10^{-2}$ in the simulations and $\sim 10^{-4}$ in the experiments (see Table 1 for simulation and experiment parameters). This analysis indicates that the behavior we observed in the experiments and simulations arises from drained layer dynamics.

### Liquefaction indicators in drained layers

Simulations and experiments determined to be controlled by drained dynamics show four indicators that are widely associated with soil liquefaction in the field and the lab: pore pressure rise, soil settlement, attenuation of shear waves, and degradation of shear modulus.

The dynamic pore pressure rises quickly in response to the onset of horizontal shaking and reaches approximately the value of the initial effective vertical solid stress (Fig. 2a). The duration at which the pore pressure remains elevated is a function of depth[27] and is set by the compaction front arrival. Once the front passes a certain depth, the pore pressure starts to decrease, so the closer a point is to the surface, the longer the pressure remains elevated at that point. The event ends at a time, $t_e$, which corresponds to the time it takes to initiate liquefaction, $t_i$, plus the time it takes the compaction front to propagate a distance $L \leq H$, from its initiation depth (Supplementary Fig. 1) to the surface, $t_e = t_i + L/u_{\text{front}}$. The initiation time, $t_i$, is found here to be exceedingly short, with a conservative median value of 0.25 s in simulations and 2.5 s in experiments (see also Section "Evaluating the

**Table 1 | Physical parameters in simulations and experiments**

| Parameter | Simulations | Experiments | Units |
|---|---|---|---|
| Mean grain density ($\rho_s$) | 2640 | 2650 | kg m$^{-3}$ |
| Fluid density ($\rho_f$) | 1000 | ~1000 | kg m$^{-3}$ |
| Mean grain radius ($r_s$) | 0.5 | 0.01 | cm |
| Fluid compressibility ($\beta_f$) | $4.5 \cdot 10^{-10}$ | ~$4.5 \cdot 10^{-10}$ | Pa$^{-1}$ |
| Fluid dynamic viscosity ($\eta$) | $10^{-3}$ | ~$10^{-3}$ | Pa s |
| Mean initial porosity ($\phi_0$) | 0.4337 | ~0.4 (mean) | – |
| Characteristic permeability ($\kappa_0$) | $6.6 \cdot 10^{-11}$ | ~$6.6 \cdot 10^{-12}$ | m$^2$ |

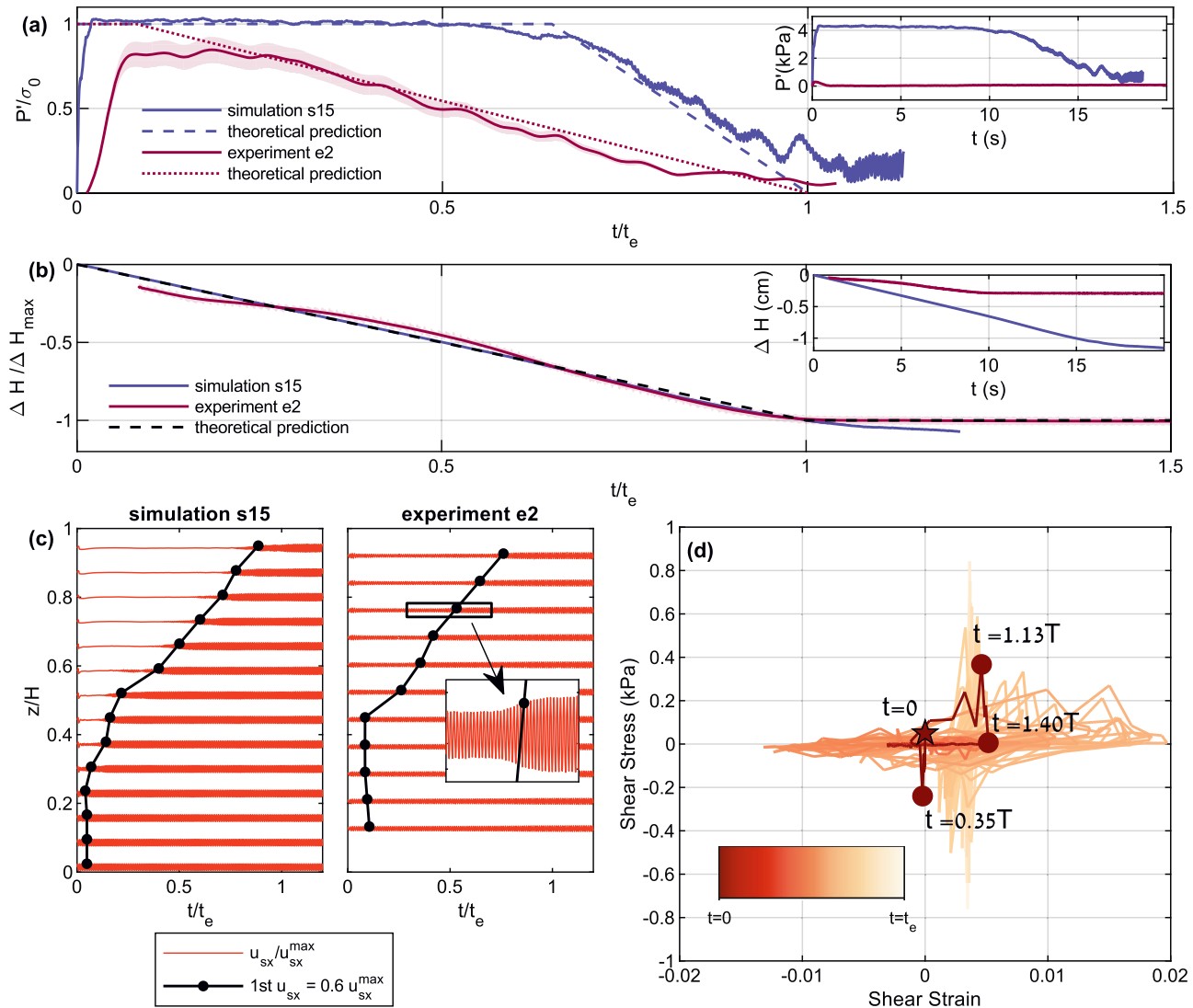

**Fig. 2 | Liquefaction indicators in drained simulations and experiments.**
**a** Dynamic pore pressure at an approximately mid-depth of the grain layer. The axes are normalized to facilitate comparison between the simulation and experiment. $t_e$ is defined based on panel b, as the time at which soil compaction significantly slows. Dashed and dotted lines represent theoretical predictions under the assumption of an infinitely narrow compaction front (Eq. (27) in ref. 27), for the simulation and experiment, respectively. The shaded red background represents the uncertainty on the experimental pressure measurement. The pore pressure starts decreasing when the front passes past the measurement depth. Inset shows non-normalized values. **b** Grain settlement and whole layer compaction. $\Delta H_{max}$ refers to the end of the linear settlement phase. Dashed line depicts the theoretical prediction based on a time integral of Eq. (6). The uncertainty on the settlement measurement in the experiment is so small as to be represented by the line thickness. The inset shows non-normalized values. **c** Shear wave attenuation. The red velocigrams represent the grains' mean horizontal velocity at various depths, normalized by the maximum value. The black lines depict the first appearance of $u_{sx} = 0.6u_{sx}^{max}$, approximating the arrival of the compaction front. Inset shows a vertical exaggeration of the black rectangle. The lower horizontal velocity, seen before front arrival, indicates a liquefied region which is unable to transmit shear waves. **d** Shear stress-strain curves in simulation s15. The color code corresponds to time in the simulations (the star marks $t = 0$). The slope of the stress-strain curves represents the shear modulus. The shear modulus degrades rapidly, within $\approx 1.4T$, where $T$ is the shaking periodicity, and then gradually strengthens.

compaction front velocity ($u_{front}$), the duration of liquefaction event ($t_e$) and the surface settlement ($\Delta H$)".

Concurrently with the pore pressure rise, the excited soil layer compacts continuously and linearly (Fig. 2b). Despite continued shaking, we find that the soil stops settling and reaches a new equilibrium configuration after time $t_e$.

Grains are shaken horizontally by the shear waves propagating from the excited layer base. Figure 2c presents the mean horizontal grain velocity time series at different depths. Shortly after the onset of shaking, shear wave amplitudes become strongly attenuated throughout the layer, as expected from a fluid-like medium. At any given depth, attenuation persists until the compaction front arrives, after which shear-waves resume the amplitude of the input shear. The

black lines in Fig. 2c follow the positions where the velocity amplitude increases back to >60% of the shaking velocity imposed at the bottom boundary, chosen here to depict the front position. The observed trend indicates that, similar to the pore pressure dynamics, attenuation lasts longer closer to the surface and overall continues up to ~$t_e$.

Figure 2d presents the relation between the shear stress and the shear strain at the mid-depth of a simulation layer. The mean slope of the stress-strain curve, known as the shear modulus, is used as a metric for the shear strength of a material[42,43]. We observe that the stress-strain curve flattens soon after the application of shaking, over less than two shear cycles, indicating that the saturated soil layer has dynamically lost its shear strength. The soil progressively regains its strength as the front progresses upwards, displaying a finite stress-strain slope.

**Table 2 | List of simulations**

| ID | Amplitude (cm) | frequency (Hz) | energy (J m⁻³) | power (J m⁻³s⁻¹) | PGV (m s⁻¹) | PGA/g | Liquefied |
|---|---|---|---|---|---|---|---|
| s1 | 0.0431 | 5.38 | 0.1395 | 0.751 | 0.0145 | 0.05 | Yes |
| s2 | 0.0431 | 7.61 | 0.2791 | 2.124 | 0.0206 | 0.1 | Yes |
| s3 | 0.0431 | 9.32 | 0.4186 | 3.901 | 0.0252 | 0.15 | Yes |
| s4 | 0.0431 | 10.77 | 0.5581 | 6.011 | 0.0291 | 0.2 | Yes |
| s5 | 0.0431 | 12.04 | 0.6976 | 8.399 | 0.0325 | 0.25 | Yes |
| s6 | 0.0431 | 13.19 | 0.8372 | 11.043 | 0.0356 | 0.3 | Yes |
| s7 | 0.431 | 2.41 | 2.7906 | 6.725 | 0.065 | 0.1 | Yes |
| s8 | 0.431 | 3.81 | 6.9764 | 26.580 | 0.1028 | 0.25 | Yes |
| s9 | 0.0431 | 3.81 | 0.0698 | 0.266 | 0.0103 | 0.025 | No |
| s10 | 0.0431 | 6.59 | 0.2093 | 1.379 | 0.0178 | 0.075 | Yes |
| s11 | 0.0431 | 8.51 | 0.3488 | 2.968 | 0.023 | 0.125 | Yes |
| s12 | 0.0215 | 7.61 | 0.0696 | 0.530 | 0.0103 | 0.05 | Partially |
| s13 | 0.0215 | 10.77 | 0.1392 | 1.499 | 0.0145 | 0.1 | Yes |
| s14 | 0.0215 | 13.19 | 0.2088 | 2.754 | 0.0178 | 0.15 | Yes |
| s15 | 0.0862 | 6.59 | 0.8372 | 5.517 | 0.0356 | 0.15 | Yes |
| s16 | 0.0862 | 5.38 | 0.5581 | 3.003 | 0.0291 | 0.1 | Yes |
| s17 | 0.0862 | 3.81 | 0.2791 | 1.063 | 0.0206 | 0.05 | Yes |
| s18 | 0.0862 | 2.68 | 0.1395 | 0.374 | 0.0145 | 0.025 | No |
| s19 | 0.0215 | 5.38 | 0.0348 | 0.187 | 0.0073 | 0.025 | No |
| s20 | 0.0431 | 4.49 | 0.0977 | 0.439 | 0.0122 | 0.035 | No |

**Table 3 | List of experiments**

| ID | Amplitude (cm) | frequency (Hz) | energy (J m⁻³) | power (J m⁻³s⁻¹) | PGV (m s⁻¹) | PGA/g | Height (m) | $\phi_0$ |
|---|---|---|---|---|---|---|---|---|
| e1 | 0.0789 | 10 | 1.6293 | 16.293 | 0.0496 | 0.318 | 0.106 | 0.4 |
| e2 | 0.0413 | 10 | 0.4452 | 4.452 | 0.0259 | 0.166 | 0.072 | 0.42 |
| e3 | 0.0733 | 10 | 1.4054 | 14.054 | 0.0461 | 0.295 | 0.07 | 0.4 |
| e4 | 0.0491 | 10 | 0.6313 | 6.313 | 0.0309 | 0.198 | 0.077 | 0.36 |
| e5 | 0.0767 | 14 | 3.0158 | 42.221 | 0.0675 | 0.605 | 0.074 | 0.41 |
| e6 | 0.0762 | 14 | 2.9728 | 41.619 | 0.067 | 0.601 | 0.071 | 0.4 |
| e7 | 0.0517 | 14 | 1.3714 | 19.200 | 0.0455 | 0.408 | 0.075 | 0.4 |
| e8 | 0.026 | 10 | 0.1763 | 1.763 | 0.0163 | 0.104 | 0.07 | 0.43 |
| e9 | 0.0242 | 10 | 0.1536 | 1.536 | 0.0152 | 0.098 | 0.073 | 0.4 |

## Drained liquefaction beyond the near field in simulations and experiments

The simulations and experiments were forced with a range of shaking amplitudes ($A$) and angular frequencies ($\omega$) (Tables 2 and 3), leading to an energy density range of 0.07–7 J m⁻³. The average seismic energy density in one shear cycle is calculated as[16,44] $e = (\rho_s/4)\text{PGV}^2$, where PGV $= A\omega$ is the amplitude of the imposed harmonic cyclic velocity. Thus, the four liquefaction indicators described above emerged although the input energy density corresponded to low, far-field, values and was smaller than the previously established liquefaction triggering threshold of 30 J m⁻³ [15,16].

Analysis of simulation results further shows that the change in porosity across the compaction front, $\Delta\phi = \phi_0 - \phi_c$, correlates with the applied seismic energy density (Fig. 3a), and has an even better correlation with the rate of the seismic energy density input (Fig. 3b), which can be evaluated as the seismic energy density over one period of shaking, $e/T$ (or as the seismic power proportional to PGV·PGA in mono frequency harmonic oscillations).

Consistent with the prediction of Eq. (4), we further find that a larger porosity change is associated with a slower propagating compaction front (Fig. 3c) and a longer liquefaction event, $t_e \propto 1/u_{\text{front}}$.

Combining the above dependencies (Fig. 3a, c), a power-law relation emerges between the front velocity scaled by the permeability and the energy density $(\log_{10}[e] = 9.42 - 0.6\log_{10}[u_{\text{front}}(\kappa(1 - \phi_0))^{-1}]$; Fig. 3d). Forcing the system with a large energy density, yet lower than the previously predicted liquefaction triggering threshold based on undrained consolidation, generates more compaction, a slower front velocity, and a longer liquefaction event. In contrast, a small energy density input induces only a small change in porosity across the front, leading to a rapid front propagation and a short-lived liquefaction event.

The co-seismic dependency of the compaction front velocity and the amount of compaction on the shaking characteristics (seismic power), differ from the often observed "solidification front"[31,38], where a front controls post-seismic consolidation, following undrained liquefaction. In contrast to such post-seismic, post-liquefaction, compaction, in drained liquefaction, the very migration of the co-seismic compaction front is the source of the drained liquefaction above it.

## The dynamics of drained liquefaction

The four liquefaction indicators observed in the low De number simulations and experiments demonstrate that liquefaction can initiate under drained conditions. In such cases, efficient drainage is key in

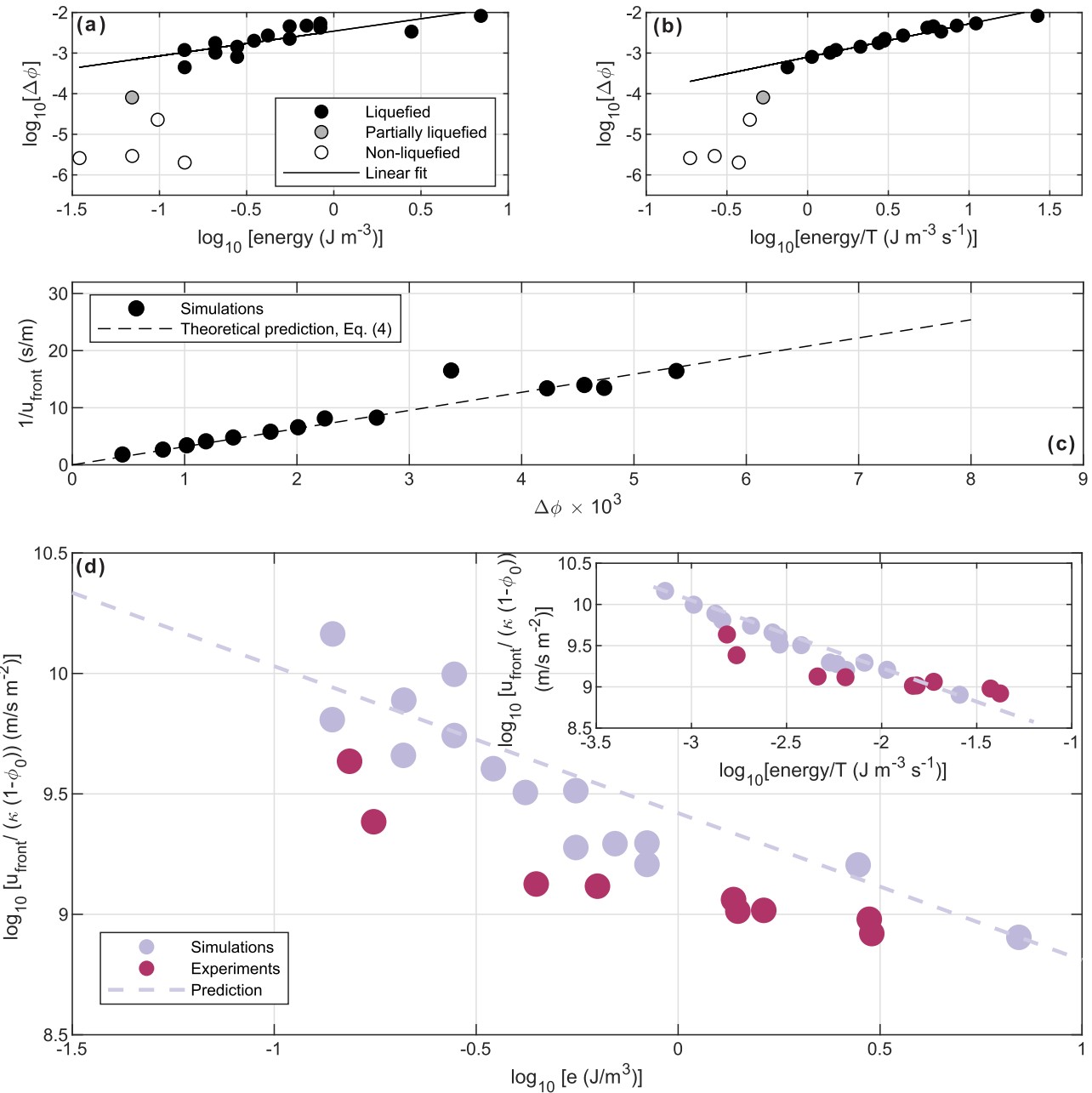

**Fig. 3 | Relations between porosity change, seismic energy density, rate of energy density and compaction front velocity. a** Change in porosity $\Delta\phi$ across the front vs. the imposed seismic energy density in the simulations. A linear fit is depicted by the solid black line ($Y = 0.61X - 2.46$; $R^2 = 0.66$). Gray and white markers were excluded from the linear fit. **b** Change in porosity vs. the rate of seismic energy density input (seismic power). The linear fit is depicted by the solid black line ($Y = 0.82X - 3.1$; $R^2 = 0.95$). **c** Inverse compaction front mean velocity vs. $\Delta\phi$. The black dashed line is the theoretical prediction of Eq. (4). **d** Normalized front velocity vs. the imposed seismic energy density in simulations and experiments. The inset shows the normalized front velocity vs. the rate of the seismic energy density input

(seismic power). Note that the energy densities (abscissa) used in the simulations and experiments, which showed the four liquefaction indicators, are below the undrained liquefaction triggering threshold of 30 J m$^{-3}$. The emerging trend shows that the compaction front propagates faster, and the duration of the liquefaction event, $t_e$, is shorter, when the energy density (and seismic power) are lower. The prediction for the simulations (dashed line with a slope of −0.6) is based on Eq. (4) and the linear fit presented in Fig. 3a (Fig. 3b for the inset). The experimental data show a similar power-law exponent (slope). The errors on the normalized front velocity are smaller than symbol size.

facilitating the dynamic rheological change of the soil layer. The upward fluid flow between the compaction front and the free surface generated lithostatic pore pressure gradients (Fig. 2a) that supported the weight of the grains, so that granular contact forces vanished. The loss of grain contacts caused the shear modulus to drop (Fig. 2d) and the shear waves to attenuate (Fig. 2c). Ongoing evacuation of fluid from the compaction front facilitated homogeneous continuous layer settlement (Fig. 2b).

Relying on the inferred drained conditions (De ≪ 1), Eq. (2), the description of the pore pressure evolution can be simplified by neglecting the first term relative to the second and third terms. Consequently, it reduces to a two terms equation[27]:

$$u_{sz} = \frac{\kappa_0}{\eta}\frac{\partial P'}{\partial z}, \qquad (5)$$

where $u_{sz}$ is the downward solid grains velocity. Equation (5) describes a compaction-pressurization feedback whereby the pressure gradient responds to instantaneous grain velocity and holds no memory of the previous pressure state[22]. In accordance with the prediction of the compaction front model, below the compaction front, where $u_{sz} \approx 0$, no dynamic pressure gradient develops and the total pressure gradient is approximately hydrostatic. Above the front, the grains settle at a uniform velocity, leading to a uniform pressure gradient. The compaction front coincides with the location where $u_{sz}$ changes from finite to zero (Fig. 1b and insets).

Similar to the undrained end-member, drained liquefaction is triggered by shaking-induced destabilization of the granular skeleton through sliding and rolling over grain contacts. At the lowest position of failure, compaction occurs relative to the stable grains below, potentially prescribing the initiation depth of the compaction front, $z_{\text{front}}(t_i)$ (see Supplementary Note 3 for more details). The pressure gradient and seepage forces that develop in response to this initial compaction only partially support the weight of the settling grains. As long as the pressure gradient remains smaller than lithostatic, the force balance on the settling grains promotes downward acceleration and faster settlement, leading to greater pressure gradients. Once the pressure gradient reaches lithostatic values, it fully supports the weight of the grains. The force balance over the settling grains is then zero, and the grains continue to settle at a constant, terminal velocity[27]:

$$u_{sz_C} = \frac{\kappa_0}{\eta}\frac{d\sigma_0}{dz},\qquad(6)$$

where subscript $C$ stands for a constant velocity. This terminal constant velocity dictates the linear compaction trend observed in Fig. 2b.

The timescale associated with pore pressure rise to lithostatic values in the simulations and experiments is short, and likely related to a rapid downward propagating liquefaction front. Such a behavior was previously identified in experiments as an unloading front[38,45,46] (see Methods Section "Evaluating the compaction front velocity ($u_{\text{front}}$), the duration of liquefaction event ($t_e$) and the surface settlement ($\Delta H$)" and Supplementary Note 6). We observed that the deepest location to which the down-going unloading front reaches correlates with the imposed shaking frequency (Supplementary Note 3), and that the unloading front reaches this deepest location in less than two shear cycles, consistent with previous experiments conducted under drained conditions[31,32]. Notably, our theory does not predict a depth, or normal stress limit for the depth, at which drained liquefaction could be triggered, and similar dynamics at much greater depths[25,39] can be identified (see Supplementary Note 5 for details). Another timescale operating in the system is the time required for an isolated, fully immersed, grain to reach its terminal downward velocity. However, since this timescale is exceedingly small, $10^{-3}$–$10^{-8}$ s (see Supplementary Note 2), the acceleration of a single grain is not a rate-limiting process for triggering drained liquefaction. Recent cyclic triaxial experiments[28] found that the number of cycles required to initiate liquefaction under drained conditions is smaller than under undrained conditions, supporting the hypothesis that a pressurization time of the order of a few cycles could be indicative of drained liquefaction initiation. Such a consideration might apply to a recent ground motion analysis showing that, in some cases, the time for liquefaction triggering is as short as ~1.7 s from the onset of recorded earthquake shaking[47,48].

Different approaches have been proposed to evaluate soil liquefaction potential. Among these, the shear stress or the earthquake peak ground acceleration (PGA)[26] forms the theoretical basis for the widely used "simplified procedure for evaluating soil liquefaction potential"[49–51]. Other approaches emphasize the shear strain[14] or the seismic energy[15,52–54] in identifying liquefaction triggering thresholds. Importantly, although the three approaches are mechanically linked, their predictions could differ[55]. The current numerical liquefaction events show a good correlation between the seismic energy density (which follows PGV[2]) and settlement magnitude (Fig. 3a). Furthermore, both numerical and experimental results show a good correlation between the seismic energy density and the front velocity (Fig. 3d). An even better correlation is found with the rate of seismic energy density input (seismic power, Fig. 3b, d). Both robust correlations emerged despite inherent differences in the boundary conditions, geometrical setup, and particle shapes between the simulations and experiments, suggesting that, within the framework of drained liquefaction triggering, the seismic energy density, and possibly a new measure, the rate of seismic energy density input (seismic power), can be considered as controlling parameters on the magnitude and duration of liquefaction events.

A leading energy-based approach for evaluating soil liquefaction potential uses the earthquake's Arias intensity[54]. While the Arias intensity is a cumulative measure that accounts for the amplitude and frequency content throughout the duration of the earthquake, the rate of seismic energy density input ($e/T$) considered here, can be interpreted as a quasi-instantaneous Arias intensity or an average power of ground shaking over one shear cycle. The excellent performance of the seismic power in explaining the amount of compaction (Fig. 3b) and the front velocity (Fig. 3d inset), and in defining the clearest threshold between liquefied and non-liquefied simulations (Fig. 3b) suggests that the drained liquefaction dynamics depends on the momentary power rather than on the cumulative power. This is likely in contrast to undrained liquefaction, which is a cumulative process by nature (the volumetric strain required to initiate liquefaction is accumulated over many shear cycles[13]), hence it might depend on a cumulative energy measure like Arias intensity.

## Drained liquefaction beyond the near field in nature

Our simulations and experiments show that drained liquefaction (with De $\ll 1$) is triggered when forced with an energy density < 30 J m$^{-3}$ and as small as 0.1 J m$^{-3}$. In natural settings, as well, the De number (Eq. (3)) can be evaluated to be smaller than one. For example, using representative values of a 5 m deep soil layer, comprising 1 mm diameter grains, and assuming a 20 grains thick compaction front, gives De $= 10^{-4} - 10^{-1}$, when the permeability range is $\kappa_0 = 10^{-9} - 10^{-12}$ m$^2$. Consequently, drained liquefaction initiation can be invoked as a general mechanism to explain field observations of liquefaction beyond the earthquake near field, accounting for the previously puzzling 61% of the events reviewed in refs. 16,17 (Fig. 4a).

We propose that the compaction-pressurization feedback, inherent to the drained compaction front dynamics[27], is a pivotal player in neutralizing the energy density threshold. With this feedback, small compaction induced by low energy density[56] (or more precisely, low rate of energy density input, $e/T$), presumably facilitated by failure of the weakest grain contacts[57,58], generates the initial pressure gradient. The associated seepage forces partially support the weight of the surrounding grains, weakening their contacts and promoting further sliding between grains, compaction, and pressurization, until a lithostatic pressure gradient is achieved and complete liquefaction occurs.

The data of field liquefaction events[16,17] show that the number of recorded events decays relatively rapidly below $e = 1$ J m$^{-3}$ and no events are recorded when $e < 0.1$ J m$^{-3}$ (Fig. 4a). Others[59] observed a similar trend regarding the decay of field liquefaction events with low PGV (proportional to the square root of the seismic energy density[44]), where no liquefaction was observed below PGV $= 0.03$ ms$^{-1}$ ($e \approx 0.5$ J m$^{-3}$). The control of the input energy density and the layer permeability, on the compaction front velocity, can explain these observations: With a lower energy density or a larger permeability, the front velocity increases (Eq. (4)), producing a short-lived liquefaction event (Fig. 4b), that is less likely to be observed or recorded. Furthermore, since under a lower energy density, compaction across the

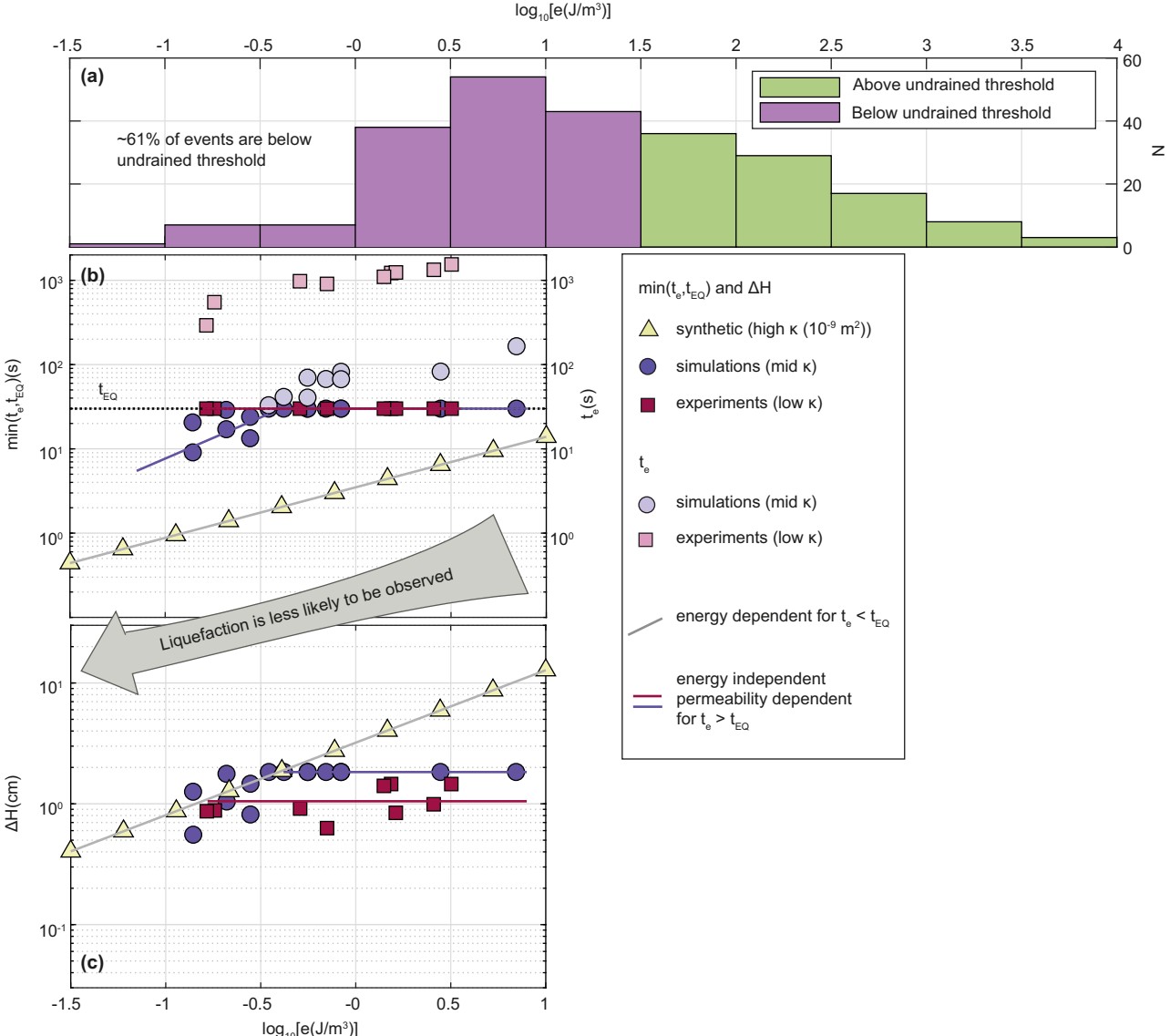

**Fig. 4 | The duration and magnitude of drained liquefaction explain the distribution of field data beyond the near field. a** Liquefaction data from ref. 17 classified by seismic energy density, *e*, show 61% of documented field events occur below the undrained liquefaction threshold and are therefore unexplained by the undrained process. In contrast, the numerical and experimental liquefaction events depicted in panels **b** and **c**, all occur below the undrained threshold. **b** Event duration, $t_e = H/u_{front}$ (see Sections "Liquefaction indicators in drained layers" and "Evaluating the compaction front velocity ($u_{front}$), the duration of liquefaction event ($t_e$) and the surface settlement ($\Delta H$)") calculated for $H = 5$ m as function of the energy density of the imposed cyclic shear. The black horizontal line depicts a typical duration of a moderate to large earthquake, $t_{EQ} = 30$ s. Light colored markers and the RHS axis show durations of the predicted events if they were

unbounded by the earthquake duration. Darker shaded markers and LHS axis present the duration of the potentially recordable liquefaction event, $\min(t_e, t_{EQ})$. Yellow markers are synthetic data with assumed higher permeability. **c** Calculated surface settlement, assuming a five-meter deep soil column ($H = 5$ m). For high permeability soils and/or low energy density, $t_e < t_{EQ}$, and compaction increases with energy density (symbols that follow the gray line). When the permeability is lower and/or energy density is higher, $t_e > t_{EQ}$, drained liquefaction ceases after $t_{EQ}$, and the total settlement is independent of the energy and depends on the permeability (symbols following the red and blue horizontal lines). Overall, liquefaction is less likely to be observed under low energy input due to its short duration and smaller surface settlement.

front, $\phi_0 - \phi_c$, is smaller (Fig. 3a), less ground settlement occurs (Fig. 4c), so that the associated hazard in such low-energy cases is potentially reduced. The decay of the number of events, and the limit on documented liquefaction event[16,17], could therefore be explained within the drained liquefaction triggering framework as a combination of an asymptotically shorter-lived (Fig. 4b) and smaller-settlement events (Fig. 4c), and a seismic energy threshold below which liquefaction does not occur (Fig. 3a, b). Alternatively, it is possible to define a PGA threshold[26,27,60] for liquefaction triggering in the current simulation set, which we find to have an exceptionally low value of $A\omega/g \approx 0.05$ (see Table 2). The numerically-identified low liquefaction

triggering threshold might be partially related to model parameters, including the use of perfectly smooth and spherical grains. More realistic grain shapes could potentially necessitate a higher triggering threshold[61]. Nevertheless, our experiments with natural sand grains showed liquefaction triggering at PGA values as low as $A\omega/g \approx 0.1$ (the precise threshold was not investigated). This suggests that the intrinsic characteristics of natural grains are unlikely to alter the outcomes significantly, permitting low PGA liquefaction triggering under real-world conditions. Furthermore, after the triggering phase, once granular contacts are minimized, the dynamics of the liquefied layer are expected to be independent of grain shape, as evidenced by the

congruence of front velocities across simulations, experimental results, and theory (depicted in Fig. 3d).

The analysis above does not account for the finite duration of earthquakes, $t_{EQ}$, which can be shorter or longer than the event termination time, $t_e$ (dictated by the front velocity, as discussed above). After time $t_{EQ}$, the seismic energy input drops to zero. The post-seismic evacuation of the excess pore pressure occurs at a rate that depends mainly on the permeability and, in our simulations and experiments, is accompanied by negligible residual compaction. The real duration of a drained liquefaction event is thus the minimum between $t_e$ and $t_{EQ}$, where $t_{EQ}$ is typically of the order of tens of seconds for moderate to large earthquakes (Fig. 4b, c). When the permeability is high and the seismic energy low, $t_e < t_{EQ}$, the total compaction is dictated by the front passing through the whole layer, converting the porosity from $\phi_0$ to $\phi_c$, where $\Delta\phi$ is controlled by seismic energy density (sloping gray line in Fig. 4c). However, when the permeability is low, and the seismic energy is high, $t_e > t_{EQ}$, the front does not have sufficient time to sweep the whole layer during the earthquake. The total compaction, in this case, is independent of seismic energy density, and instead depends on permeability. The permeability controls the settling velocity of grains in the liquefied sub-layer (Eq. (6)), and the total compaction is then the integral of the settling velocity over time $t_{EQ}$ (horizontal lines in Fig. 4c).

Finally we discuss briefly two common scenarios of more complicated geometries: 1) when the water table does not coincide with the surface, and 2) a soil with layered permeability. In our simulations and experiments, the water table was taken to coincide with the grain layer free surface. In such settings, fluid expulsion out of the soil layer will start concurrently with drained liquefaction initiation. However, if the water table lies much below the surface, fluid expulsion out of the soil during drained liquefaction initiation could be delayed. In cases where the water table is sufficiently deep, and the energy density is relatively low (inducing only a small $\Delta\phi$), the water table might not reach the surface during liquefaction, and no fluid expulsion out of the surface would be observed. Coseismic settling of the ground surface, on the other hand, will take place even in the absence of fluid expulsion. Delayed fluid expulsion should therefore, not be a-priori considered as an indication for undrained liquefaction initiation followed by a breach of low permeability barrier, or as an indicator for liquefaction by pore pressure diffusion from a distant source[7,16,18]. Instead, it could also be consistent with a drained initiation if the water table was originally relatively deep (see Supplementary Note 4).

The next important case is when permeability is not homogeneous as in our simulations and experiments, and instead includes sub-layers with variable permeability. Centrifuge experiments[62] and numerical simulations[63] show that a water-film may form below a low permeability seam. If the fluid flow upwards across the seam is slow and the seam is not broken yet, the water-film may change its volume to preserve a constant lithostatic water pressure boundary condition for the layer below the seam (by "pushing" the seam upwards[62]). This promotes a behavior very similar to the presented drained compaction front, where the higher permeability sub-layer below the seam is compacting and the fluid drains towards the water film (rather than directly to the surface). A further investigation of such a setting is needed to examine the initial pressurization process and its sensitivity to the seismic energy density.

In conclusion, we summarize that theory, simulations, and experiments demonstrate that drained liquefaction triggering could be invoked to explain ubiquitous and previously puzzling observations of liquefaction beyond the earthquake near-field[16], where the seismic energy density is lower than an empirical threshold inferred for the onset of undrained liquefaction.

Effectively drained conditions are associated with a pore pressure diffusion timescale shorter than the grain skeleton deformation timescale, such that the non-dimensional Deborah number, De ≪ 1.

Such conditions represent combinations of site and event properties, where high permeability throughout the soil column and low shaking frequency contribute to the drained conditions. The latter is also associated with smaller energy density input. This suggests that the drained initiation end-member likely dominates far away from the earthquake epicenter (in the intermediate to far-field).

The dynamics of drained liquefaction are controlled by a co-seismic, upward-migrating compaction front that induces co-seismic, spatially heterogeneous compaction. Theoretical predictions, simulations, and experiments further reveal that the compaction front velocity inversely correlates with the seismic energy density, and shows an even better correlation with the rate of seismic energy density, a new seismic-intensity-based control parameter. Consequently, exceedingly small events with energy density $\lesssim 0.1\,\mathrm{J\,m^{-3}}$ will be characterized by a rapid compaction front and small co-seismic compaction, potentially hindering field detection of liquefaction and explaining the lack of documented liquefaction events at very low energy density[16].

A critical implication of this study is that liquefaction potential and risk evaluation should account for the possibility of drained liquefaction triggering, with its general mechanistic model for liquefaction of well-drained soils, i.e., young fluvial and beach sediments and reclaimed lands, under exceedingly small seismic energy density input.

## Methods

### A general description of the grain-fluid system

We study the coupled grain-fluid dynamics of a fully saturated granular layer subjected to 1D horizontal harmonic shear displacement. The top of the layer is unconfined. Horizontal cyclic shear is applied to the base of the layer, which acts as a no-flow boundary for the fluid. This geometry represents a shallow soil layer overlaying a bedrock that is agitated by an upward traveling horizontally polarized seismic shear wave.

### Numerical simulations

The numerical approach is described in[22,27]. Here, we repeat its main details. We use a two-phase coupled model. The grains are modeled using the discrete element method[64], and the interstitial pore fluid is modeled as a continuum on a superimposed Eulerian grid[34,40,65–68].

Grain velocity and position are resolved by time integration of the linear and rotational momentum conservation equations[27]:

$$m_i\dot{\mathbf{u}}_{s,i} = m_i\mathbf{g} - V_{\mathrm{imm},i}\rho_f\mathbf{g} + \Sigma_j\mathbf{F}_{ij} - \frac{\boldsymbol{\nabla}P' \cdot V_i}{1-\phi} \tag{7}$$

$$I_i\dot{\boldsymbol{\omega}}_{s,i} = \Sigma_j R_i\hat{\mathbf{n}}_{ij} \times \mathbf{F}_{ij}, \tag{8}$$

where $\dot{\mathbf{u}}_{s,i}$ and $\dot{\boldsymbol{\omega}}_{s,i}$ are the translational and rotational accelerations of grain $i$ (dot notation indicates time derivative) and $m_i$ and $I_i$ are the mass and moment of inertia of grain $i$. $R_i$ is the radius of grain $i$ and $\hat{\mathbf{n}}_{ij}$ is a unit vector along the direction connecting the centers of grains $i$ and $j$. In Eq. (7), the first term on the right-hand side is the gravitational force, where $\mathbf{g}$ is the gravitational acceleration. The second term is the buoyancy force, whose magnitude depends on the grain immersed volume $V_{\mathrm{imm},i}$ and the fluid density $\rho_f$[26]. The third term is the sum of contact forces ($\mathbf{F}_{ij}$) over all grains $j$ that are in contact with grain $i$, calculated with a linear contact model[64]. The fourth term represents the seepage force exerted by the gradient of the dynamic pore pressure, $\boldsymbol{\nabla}P'$, where $V_i$ is the volume of grain $i$.

The evolution of the interstitial fluid pressure is represented by Eq. (1)[22], which is solved by using an implicit scheme over a square grid, with a grid spacing of two average grain diameters[21,22,34,66]. No a-priori assumption is made regarding the value of the De number (Eq. (3)), and the full three terms equation is solved.

The two-way coupling between the grains and the fluid is implemented as follows. The fourth term on the right-hand side of Eq. (7) is evaluated via a bilinear interpolation of $\nabla P'/(1-\phi)$ from the surrounding grid nodes to grain $i$. The second and third terms of Eq. (1) are evaluated by defining smooth fields of grain velocity and porosity over the grid through a bi-linear interpolation of grain radius and velocity from individual grains surrounding each grid node. The permeability, $\kappa$, in Eq. (1) is calculated based on a three dimensional Kozeny–Carman relation[69]:

$$\kappa = \kappa_1 \kappa'(x, y, t) = \alpha r^2 \frac{\phi^3}{(1-\phi)^2}, \tag{9}$$

where $r^2$ is the bi-linearly interpolated squared grain radii in the surroundings. $\kappa_1 = \alpha <r>^2$ is a constant prefactor, and $\kappa' = r'^2 f(\phi)$ captures permeability variations in space and time. $<r>$ is the mean grain radius in the system and $r'$ is the local deviation from it, such that $r = <r>r'$. In the original Kozeny–Carman relation, $\alpha = 1/45$[69] is a geometrical prefactor for spheres. In our simulations, we vary $\alpha$ to directly control the order of magnitude of the permeability independent of the grain size[22,27].

The geometry of the numerical layer (Fig. 5a) is a Hele-Shaw cell comprising spherical grains with grain radii between 0.8–1.2 cm, drawn from a normal distribution with a mean of 1 cm and a standard deviation of 1 cm. The system's horizontal dimension is 0.4 m. The layer is prepared as follows: First, a target height is specified. Then, grains are sedimented under gravity onto the bottom wall in a fluid-free environment. Next, to slightly compact the layer, a short horizontal shaking phase is applied over 0.62 s with $f = 12$ Hz and amplitude of $A = 0.0431$ cm, followed by 0.13 s relaxation, where no external forces aside from gravity are applied. Finally, the fluid is added so its height approximately coincides with the top of the grain

layer and the layer is relaxed again. In the simulations presented here, the initial layer height, following the preparation stage, is $H \approx 1.44$ m.

The bottom wall of the numerical Hele-Shaw cell is made of half grains glued together. The boundary condition for the bottom wall is zero velocity in the vertical direction ($u_{sz}(z = \text{bottom}, t) = 0$) and sinusoidal displacement in the horizontal direction, $x(z = \text{bottom}, t) = A(1 - \cos(\omega t))$, where $A$ and $\omega$ are the shearing amplitude and angular frequency, respectively. At the top boundary, there are no normal or shear stresses. The boundary conditions for the fluid phase are no flow boundary at the bottom ($\partial P'/\partial z(z = \text{bottom}, t) = 0$) and constant pressure boundary at the top ($P'(z = \text{top}, t) = 0$). The water level is maintained at its initial height throughout the simulation. The side boundaries are periodic for the grains and pore fluid, mimicking a laterally infinitely long layer.

Table 1 summarizes the simulations' parameters. Table 2 lists the simulations presented here with their applied shear amplitude and frequency. The pressure signal in Fig. 2a is smoothed over a window of two cycles. The compaction in Fig. 2b is calculated as the time integral of the mean vertical velocity of grains in the topmost sub-layer.

### Experiments

The experiments (Fig. 5b–e) comprise a $12 \times 12 \times 12$ cm$^3$ transparent box. The box is attached to a horizontal shaker (Tira® S51120) fed with a harmonic signal from a signal generator (Agilent® 33220A) through an amplifier (BAA500). The box's face perpendicular to the shaking direction is filmed by a high-speed camera (Photron® SA5) at a rate of 250 frames per second. The frames are analyzed by using MATLAB® image processing toolbox and PIVlab[70,71], an open-source MATLAB® toolbox, to identify changes in the layer's height and define instantaneous grain velocity. An array of three pressure transducers

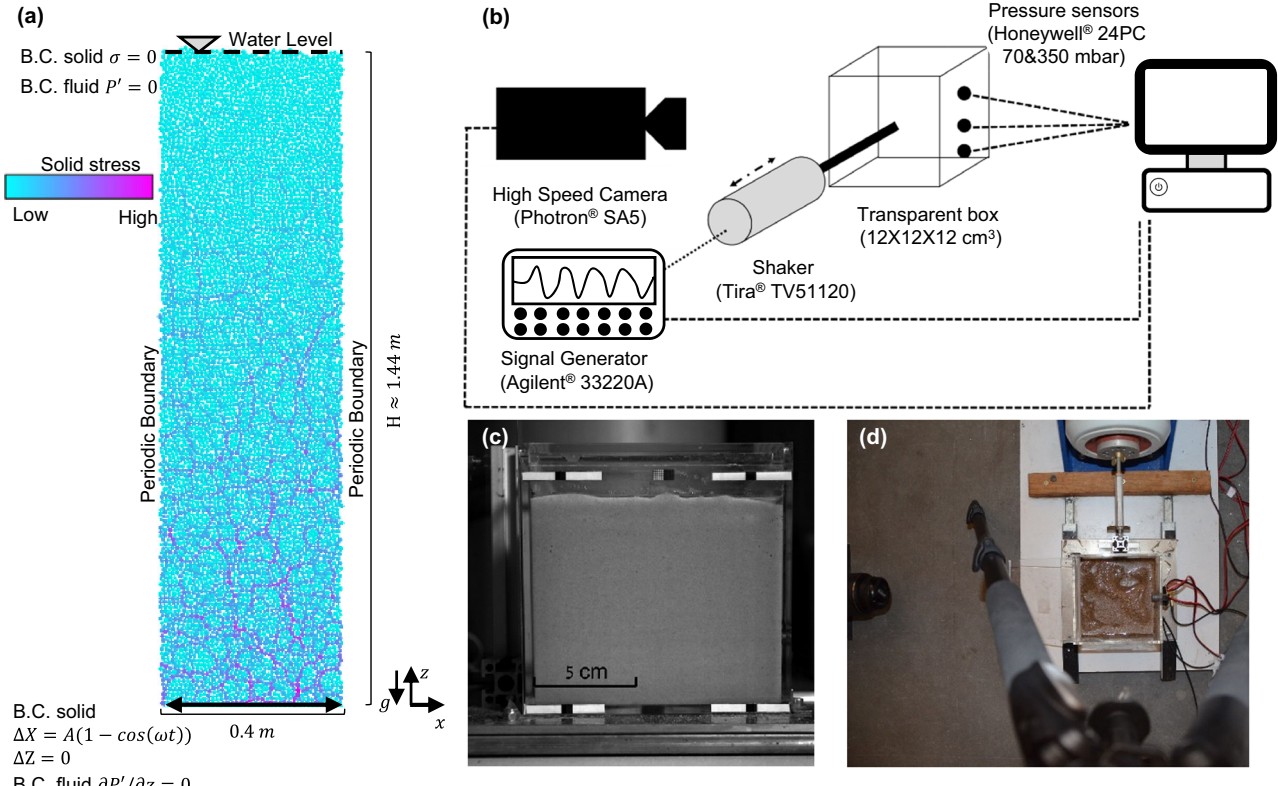

**Fig. 5 | Numerical and experimental setups. a** Schematics of the simulations setup and boundary conditions. **b** Schematics of the experimental setup. **c** The front face of the experiment box (a frame from the high-speed camera used for velocimetry).
**d** Top view of the experiment box. The shaker is seen at the top of the photo, the pressure transducers are at the right, and the high-speed camera lenses are at the left.

(Honeywell 24PC) is mounted vertically on the opposite parallel face of the box at depths of 1, 3.6, and 6.1 cm above the box base.

Before the experiment starts, the experimental box is filled with tap water, and the pressure transducers are calibrated under hydrostatic conditions. Then, sand grains with a mean diameter of 200 micrometers (SIFRACO N34) and density $\rho_s = 2650$ kg/m$^3$ are gradually poured into the box. We aim for a situation where the water table and the top grains approximately coincide. Horizontal shaking is applied for 30 s with a displacement amplitude of at least one mean grain diameter. The pore pressure is measured at a frequency of $10^4$ Hz from 30 s before the application of shaking and until 240 s after shaking stops. The pressure at the top is atmospheric, such that the top boundary is fully drained. All the other box faces exert no flow conditions.

Table 1 summarizes the experiments' parameters. Table 3 lists the experiments presented here with their applied shear amplitude and frequency. The initial porosity, $\phi_0$, presented in Table 3 is evaluated as follows: First, the pore water volume is evaluated as the difference between the water volume used in the experiment and the volume of the thin water film above the grains. Then, the pore water volume is divided by the total volume of the saturated grains based on the height of the grain layer as recorded by the first high-speed camera image.

The mean value of the pre-shaking pressure measurements is used to determine the hydrostatic pressure reference. The pressure signal in Fig. 2a is filtered using a low pass filter with a cutoff frequency of 1 Hz. The shaded red area in Fig. 2a, representing the uncertainty on the pressure, follows the 95% confidence bounds on the parameters of the linear regression between voltage and pressure based on the pressure transducers calibration stage.

The calculation of the normalized compaction in Fig. 2b is based on an edge detection algorithm that identifies the top boundary (edge) of the grain layer in individual images. The algorithm was executed several times while varying the top boundary of the search frame within which the algorithm searches for the edge. From search frame 211 (corresponding to a scaled time of $t/t_e = 0.084$) and on, the edge becomes independent of the search frame height, and thus Fig. 2b (red curve) shows the compaction trend only from frame 221. The red curve represents the normalized mean edge topography within the search frame (smoothed by a moving average window of 0.8 s). The shaded red band represents the uncertainty on the normalized edge height based on the standard error of the edge topography. The mean and standard error of the edge topography in the first search frame are based on a manual edge extraction.

The instantaneous grain velocity field is measured in every frame by using PIVlab[70,71], which relies on sub-frame correlation between timely-adjacent frames. The vertical velocity is then averaged over sub-layers, yielding the vertical velocity of grains as a function of depth and time (see Section "Evaluating the compaction front velocity ($u_{front}$), the duration of liquefaction event ($t_e$) and the surface settlement ($\Delta H$)"). To minimize boundary effects from the box's walls, the averaging is done only close to the box's center (approximately in the middle 2/4 of the box's total width).

The energy, $e$, in Fig. 4c, is based on an estimation of the imposed PGV in the experiments. The input shaking frequency was accurately controlled by setting the frequency of the shaker. The shaking amplitude was estimated based on four markers placed close to the corners of the experiment box. Markers' position was traced across frames. The temporal mean of the markers position was subtracted from the position time series of each marker, and the four position time series were averaged. Then, the peaks of the combined, averaged time series were extracted, and the shaking amplitude was estimated as the average over the absolute value of the peaks through time $t = [0, t_e]$. The uncertainty in evaluating PGV is related to the standard error of the absolute value of the peaks time series. The error propagated to $\log_{10}[e]$ is smaller than the symbol size in Fig. 4c.

The permeability in the experiments was evaluated based on five static permeability tests. A constant head was applied across a saturated sand layer in each test, prepared similarly and with the same geometry as the shaking experiments. The outlet point that was located 1.4 cm above the base of the box imposed a 3D porous flow field in the box. The cumulative outflow was measured through time, and its time derivative was used as the discharge (with units of $m^3 s^{-1}$) in a 1D Darcy's law to determine the permeability. A correction factor from a true 1D porous flow to the specific 3D flow structure in these tests was derived by simulating the two geometries in COMSOL Multiphysics. For the same material permeability, the discharge in the 3D geometry was smaller by a factor of 10 with respect to its 1D counterpart. The permeability of each experiment was then estimated as being larger by a factor of 10 with respect to the measured quantity. The hydraulic head was varied between the five experiments, and the permeability used in Fig. 4c is the mean over the five measurements. The uncertainty on the permeability is evaluated as the standard error over the five permeability measurements. When propagated to the y-axis of Fig. 4c, $\log_{10}[u_{front}/(\kappa(1 - \phi_0))]$, the uncertainty is smaller than the symbol size.

### Evaluating the compaction front velocity ($u_{front}$), the duration of liquefaction event ($t_e$) and the surface settlement ($\Delta H$)

The compaction front velocity is defined based on the ratio between the horizontally averaged vertical grain velocity and the grain terminal velocity defined by Eq. (6). Averaging is performed over sub-layers of two average grain diameter thickness. The averaged velocity is smoothed in time using a running average window of -0.67 s in the simulations and -1.2 s in the experiments. In the simulations, we further smooth the vertical dimension using a running average window of 10 cm. Plotting the averaged, smoothed and normalized velocity as a function of depth and time, $u_{sz}^{norm}(z,t)$ results in a map that highlights settling vs. stagnant grains (Fig. 6).

The compaction front depth at each time, $t$, is extracted by scanning $u_{sz}^{norm}(z,t)$ from the bottom upward and identifying the first depth where $u_{sz}^{norm}(z,t) \geq 0.5$ in simulations and $\geq 0.01$ in experiments. This depth is defined as the front location at time $t$, $z_{front}(t)$. Finally, we manually pick the time when the front starts migrating upward continuously ($t_i$) and the time of front arrival to the top of the layer ($t_f$). In some experiments and simulations, $z_{front}(t)$ loses its coherent slope at some stage, and we choose $t_f$ to be the last time step showing a coherent slope. The average front velocity is calculated as the average slope of the front depth-time curve between these two times.

The time of liquefaction event termination, $t_e$, used as the timescale factor in Fig. 2, is determined as the time at which the soil compaction slows down significantly (see Fig. 2b), as an approximation of $t_e = t_i + L/u_{front}$, where $L$ is the distance to the surface from $z_{front}(t = t_i)$. In most cases, $t_i$ is negligible in comparison to $L/u_{front}$ since the downward moving unloading front (Fig. 6 inset) is very fast.

The theoretical time of a liquefaction event in Fig. 4b (semi-transparent symbols) was extrapolated to a 5-m deep soil layer, based on the relationship between the measured compaction front velocity in simulations and experiments, and the input seismic energy density. The synthetic high permeability data ($\kappa = 10^{-9} m^2$) was calculated based on the power-law prediction derived from simulations (Fig. 3d and section "Drained liquefaction beyond the near field in simulations and experiments"), by substituting the numerical characteristic permeability with the synthetic permeability. The actual time of liquefaction (opaque symbols in Fig. 4b) was calculated as the minimum value between the theoretical time ($t_e$) and earthquake duration of $t_{EQ} = 30$ s. The sloping blue line is the best fit for the data points which satisfy $t_e < t_{EQ}$. The surface settlement in Fig. 4c was calculated as the integral of the settlement velocity (Eq. (6)), between $t = 0$ and $t = \min(t_e, t_{EQ})$. The blue and red horizontal trend lines represent the mean value of data points associated with $t_e > t_{EQ}$. The gray sloping line represents the

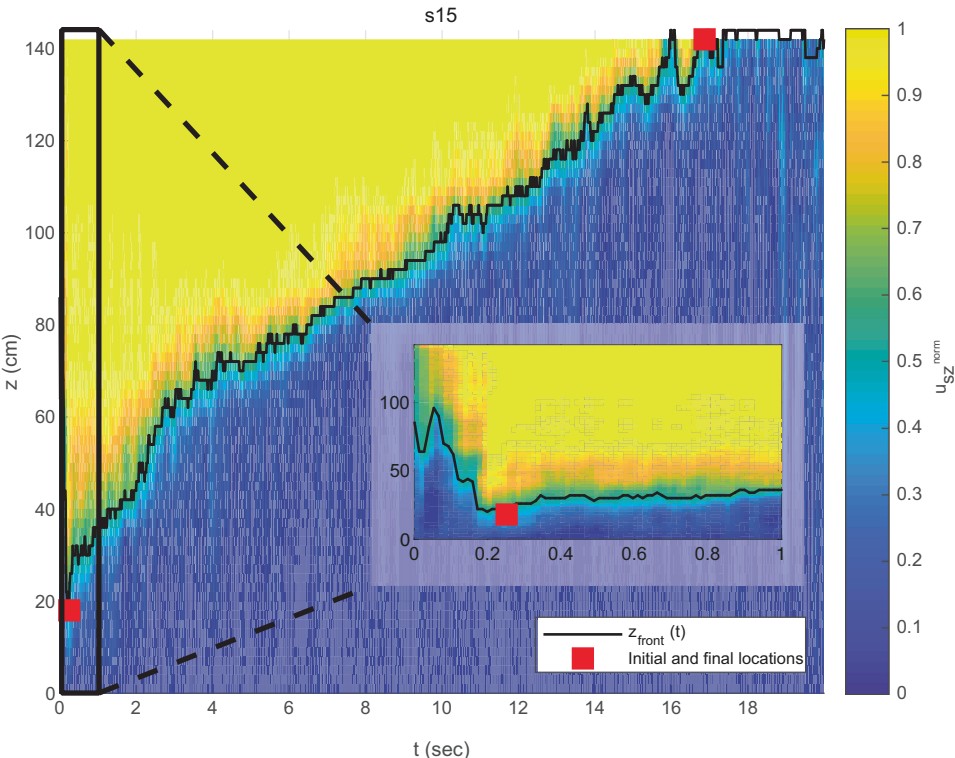

**Fig. 6 | Example of compaction front detection and calculation of the average front velocity $u_{front}$.** The vertical axis represents height above the bottom wall, and the horizontal axis is time. The color map shows the normalized averaged and smoothed vertical grain velocity, $u_{sz}^{norm}$. The black curve depicts the inferred front location ($z_{front}(t)$). The front velocity is calculated as the average slope of the black curve, between the manually picked initial and final times (red squares). The inset shows the first second of the simulation (corresponding to the black rectangle in the main panel). The downward-moving unloading front is observed in the first $\simeq 0.2$ s.

amount of settlement resulting from the synthetically adapted power-law prediction, as described above.

## Data availability
Data of simulations and experiments can be found in https://doi.org/10.17605/OSF.IO/ZFNH9.

## Code availability
Matlab code can be found in https://doi.org/10.17605/OSF.IO/ZFNH9. Further requests should be addressed to S.B.-Z.

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

## Acknowledgements
S.B.-Z. and R.T. thank Alain Steyer and Miloud Talib for technical assistance with the experiments, Martine Trautmann for granulometry and the support of Institut Français d'Israël and Campus France via the Chateaubriand Fellowship. S.B.-Z and R.T. wish to thank Yossef Hatozr and Valérie Vidal for fruitful discussions. S.B.-Z. and E.A. wish to thank Assaf Klar and Eitan Cohen for a fruitful discussion. R.T. wishes to thank Eirik Flekkøy for fruitful discussions. L.G. acknowledges support from ISF grant 562/19. E.A. thanks ISF grant #910/17. R.T. acknowledges the support of the CNRS INSU ALÉAS and CESSUR programs, the CNRS MITI program, the Universities of Strasbourg and Oslo and the Research Council of Norway through its Centre of Excellence funding scheme, project number 262644.

## Author contributions
S.B.-Z. conducted the experiments under the supervision of R.T., and the numerical simulations under the supervision of L.G. and E.A. R.T. conducted the COMSOL simulations. S.B.-Z. produced the figures with the aid of L.G.. S.B.-Z. led the writing and all authors contributed to writing and editing the manuscript.

## Competing interests
The authors declare no competing interests.
