## [Peer Review File · Nature Communications]

Drainage explains soil liquefaction beyond the earthquake near-fieldREVIEWER COMMENTS

Reviewer #1 (Remarks to the Author):

Seismic liquefaction of sand is a quite complex phenomenon. Its generation, development and dissipation are mainly restricted by the physical properties, stress state and boundary conditions of soil. There are many influencing factors, such as soil density, soil structure, clay content, saturation, gradation, water permeability, initial stress state and dynamic load characteristics. Generally speaking, the mechanism of earthquake liquefaction can be expressed as follows :

The shear wave propagates upward from the underlying rock and soil layer during the earthquake, and causes alternating stress in the soil, resulting in vibration pore water pressure, which is the main reason for the liquefaction of saturated sand. Under the action of alternating stress, new stress will be generated at the contact point of soil particles. When this stress reaches a certain value, the original connection and structural state between soil particles will be destroyed, and the sand particles will be separated from each other. At this time, the stress originally transmitted by sand particles through the contact point will be transmitted to the water in the pores to bear, resulting in the increase of pore water pressure. With the increase of the number of stress cycles, the pore water pressure increases due to the gradual accumulation. On the one hand, pore water tries to discharge upward under a certain vibration pore water pressure; On the other hand, soil particles try to sink down under the action of self weight, so that at the moment of structural damage or within a certain period of time, the downward sinking of soil particles is hindered by the upward discharge of pore water, so that the soil particles are partially or completely suspended, and the shear strength disappears locally or completely, which means that the soil has varying degrees of deformation or complete liquefaction.

In other words, the seismic liquefaction of sand is related to the ground motion excitation and the properties of soil. The ground motion excitation here mainly refers to the peak acceleration and the duration of earthquake. For the earthquake liquefaction phenomenon in the region far away from the epicenter mentioned by the authors, being far away from the epicenter does not mean that liquefaction will not occur, and the peak acceleration in the region far away from the epicenter may not be low. Studies have found that liquefaction occurs when the earthquake intensity is as small as moment magnitude 4.5 (Green and Bommer, 2020). The existing mechanism of seismic liquefaction mentioned by the authors is based on undrained conditions. In fact, drainage and undrained conditions are not unnecessary conditions for seismic liquefaction of sand. Undrained conditions are only to explain that pore water pressure offsets gravity. Even under drainage conditions, pore water pressure may dissipate slowly, but sand liquefaction is slow. Soil structure is also an important factor to control the liquefaction of sand. After the structure is damaged by earthquake vibration, the vibration pore water pressure gradually increases with the increase of the number of stress cycles, and its size is finally enough to make the saturated sand lose its shear capacity locally or completely. That is, the seismic liquefaction of sand is affected by the compactness of soil. Even if there is a new mechanism of sand seismic liquefaction, it should be based on particle mechanics, such as the so-called shear jamming mechanism. The innovation of this study is limited.

In general, according to the current status of this study, the reviewer regrets that, it is not recommended to be published in this professional journal.

Reviewer #2 (Remarks to the Author):

The manuscript titled "Drainage explains soil liquefaction beyond the earthquake near-field" examines the phenomenon of earthquake-induced liquefaction, focusing on the importance of co-seismic drainage, with implications on the conventional view of liquefaction triggering.

Methodologically, it combines analytical with numerical and experimental work. It is a very welcome addition to the growing body of work that challenges the traditional view of liquefaction as an undrained phenomenon. The results reported are noteworthy and I expect them to be of significance to the field. The work presented supports the conclusions and claims. I would recommend the manuscript for publication subject to the following points being addressed:

1. Line 109: the description of an upwards moving "compaction" front – referred to in literature also as a "solidification" front – which separates the layer into two parts, a compacted, or solidified, below the front and an uncompacted, or unsolidified, agrees with literature (also with references 28, 33, 34), though it is often referring to post-liquefaction reconsolidation, following an undrained assumption for excess pore water pressure generation. Following your compaction front analysis, how can the initial, downwards propagation of a front separating liquefied from non-liquefied soil be explained (e.g. [33])? Is this an aspect of response that can be captured with your analytical approach? This downwards propagation of a "liquefaction" front precedes the reconsolidation process, where an upwards moving "solidification" (or here "compaction") front is observed – this reconsolidation process seems to be explained well with your compaction front analysis and indeed experimental observation shows that reconsolidation typically starts during the earthquake and not necessarily after its end, agreeing with your view of liquefaction as a "drained" phenomenon.
2. Some further explanation on when and where the compaction front is formed when applying your analytical (or theoretical) approach would be welcome. In some cases (e.g. Fig2c, Experiment e2) only the top part of a deposit liquefies. Is it possible to apply your analytical approach to predict the greatest depth where soil is liquefied, and when this happens, or is this taken as a given and the compaction front analysis starts from there?
3. Line 112: The assumption of stationary particles below the compaction front is essentially the assumption of Florin and Ivanov [33] but experimental data from centrifuge testing (e.g. [28], [29]) has shown that the pore water gradient below the front is not actually hydrostatic, so that further consolidation does take place below the "compaction" front, meaning that compaction and fluid flow would still take place in region 1 of Figure 1. In any case, even if the pore water gradient were hydrostatic, some compaction is inevitable as the effective stress below the compaction front gradually increases with the upwards propagation of the front. Though it is understood that some simplification is necessary, a comment to clarify to the reader the potential limitations of the assumption made would be welcome.
4. Figure 3b: is the equation plotted with a black line eq3 or eq4?
5. References [15] and [16] are cited regarding the lower bound of energy density required to cause liquefaction. Cases below 30 J m^{-3} are given the term "unexplained". Based on this assessment, 61% of events examined in figure 4 are unexplained – an alarmingly high number. However, in practice the most common methodology to assess liquefaction triggering is the stress based simplified approach, introduced by Seed & Idriss (1967). See Boulanger & Idriss (2016) for a recent iteration of the method. This method gives a factor of safety for liquefaction occurrence. Would 61% of the events you examined remain "unexplained" if the simplified method was used? It would be very useful for the reader to know the number of "unexplained" events based on the methodology that is typically used in practice.
Seed, H. B. & Idriss, L. M. (1967). Analysis of Soil Liquefaction: Niigata Earthquake. Journal of the Soil Mechanics and Foundations Division 93, No. SM3, 83–108.
Boulanger, R. W. & Idriss, I. M. (2016). CPT-Based Liquefaction Triggering Procedure. Journal of Geotechnical and Geoenvironmental Engineering 142, No. 2, 04015065
6. Line 253-254: Some clarification is needed in this phrase to avoid confusion. You state that no further significant compaction is accumulated after the end of the earthquake. You should clarify if you are referring to the whole liquefiable deposit and if so, how you explain significant post-earthquake settlements observed during post-earthquake reconsolidation e.g. [33], [28], [29].
7. Line 357: Typically in liquefaction experiments, laminar boxes are used, where each horizontal lamination can move relative to the others. This would be closer to the periodic boundary used for the numerical simulation. Given the practically rigid box used and the small dimensions compared to similar experiments in literature (as already cited in this manuscript), a comment on potential boundary effects is necessary. Moreover, it is implied that the input motion was harmonic but it would be better if this was clearly stated in the text.

Reviewer #3 (Remarks to the Author):

This is an interesting paper that aims to explain the soil liquefaction in the far-field where the earthquake shaking intensity is small. It is true that such events have been observed in previous earthquake events and reported in case histories. Several theories exist for that such as soil

amplification in some regions of the far-field that cause liquefaction while attenuation of accelerations in other regions may not lead to full liquefaction.

The following comments are offered for the authors to consider.

1. It may be better to distinguish between 'liquefaction' and 'fluidisation' throughout the manuscript. I would use 'liquefaction' for excess pore pressures generated due to seismic shaking and 'fluidisation' for loss of shear strength of soil due to pore fluid migration (from lower layers towards the soil surface) and can happen into regions where shaking intensity is not high. Authors may want to clarify that when they said 'drained liquefaction' if they mean 'fluidisation'.

2. In the same vein, it may be better to distinguish clearly between the 'drained' and 'undrained' terms. In liquefaction research (see Refs 22, 24 and 29) the term 'partially drained' was introduced to demarcate this. In this paper the authors used the Debroah number (De) to indicate partially drained events.

3. The authors may want to clarify what are the time scales for the liquefaction and fluidisation events. One would expect liquefaction to happen (for loose sands) in the first few cycles of earthquake shaking. Fluidisation can happen and continue into the post seismic period.

4. The concept of 'liquefaction front' moving from soil surface towards the bedrock as the shaking starts and the 'densification front' moving from the bedrock towards the soil surface have been previously identified by many researchers (e.g. ref 33). There is a large body of dynamic centrifuge test data to show these fronts experimentally occurring in layers of $\sim 10\text{m}$ deep. The authors may want to include some of these references into the manuscript.

5. There is considerable body of research on liquefaction that establishes cyclic shear strain amplitude threshold i.e. earthquakes that cause shear strains above this threshold cause excess pore pressure generation while those below won't. I feel this is more physical as shear strains are physically linked to the movement of individual soil grains and the length parameter. Incidentally I agree with the authors argument about length scale in Sec. 3.1. This almost corresponds to Tatsuoka's observations of formation of shear bands and their thickness in sands (~ 20 particle diameters). However, I do not see why the energy argument is used in this paper. Measures like Arias intensity that depend on the PGV are global measures, where as liquefaction (in my view) depends on the relative displacement between particles. I am not talking about fluidisation here (as distinguished in point 1 above). The authors may want to argue why PGV is a good measure for liquefaction or otherwise.

6. In Sec. 4.1 the authors argue that the time it takes to reach 'full drained liquefaction' is exceedingly small i.e. $10^{-3} \sim 10^{-8}$ seconds. I presume that the authors mean that this is different from 'seismic liquefaction' and are saying fluidisation can occur much more quickly. Firstly, for seismic liquefaction, the excess pore pressure generation can take anything from 1 to 2 cycles to many cycles (~ 8 to 10 cycles) depending on the relative density of the sand. Loose sands liquefy more easily than dense sands. Secondly, the time for fluidisation surely depends on the permeability, porosity of the soil strata and pressure gradient applied which will yield much longer time scales, as pore fluid has to migrate from one region to the other?

7. I have several objections on the experimental program reported in Sec. 10. Also not sure why the authors do not use data from dynamic centrifuge tests on liquefaction.

a) The experiments are in a very small-scale model container 120 mm (cube). The boundary effects will be significant in such a small model container.

b) What is the relative density of the soil specimens? This should be included in Table 2. Porosity is not convenient for this as RD is measure of the soil state with respect to the loosest and densest possible states.

c) The frequency of shaking is very high (10 Hz, most earthquakes are in the 1 to 5 Hz range)

d) The amplitude of shaking is very small 0.8 mm is the largest amplitude used. How does this compare to the mean particle size (D_{20} or D_{50}) for this sand?

e) Referring to Fig.2a, the excess pore pressures do not show any suction cycles, cyclic oscillations (to indicate the crossing of phase transformation line (following Ishihara) nor reach full liquefaction

of 1. The authors may want to explain why.

f) The time for generation of excess pore pressure in Fig. 2a is much more slower than in the simulations and magnitudes are smaller.

g) In fig. 2d, the shear strains are biased towards the positive side? Why?

h) In Fig.2d, are the experimental results and simulations at the same frequency? The recoupling and increase in oscillations in the experiment are not clear.

Dear reviewers,

We wish to thank you for putting the time and effort in reviewing our manuscript. The comments you made are significant and helped us notice some previously hidden aspects of our work, as well as highlight several “blind spots” that we had when communicating our study.

We believe that the changes we have introduced to address your comments have greatly improved the manuscript, both scientifically and in terms of science communication. We hope that you'll see those changes similarly to how we perceive them.

Sincerely,

Shahar Ben-Zeev on behalf of the co-authors

REVIEWER COMMENTS

Reviewer #1 (Remarks to the Author):

Seismic liquefaction of sand is a quite complex phenomenon. Its generation, development and dissipation are mainly restricted by the physical properties, stress state and boundary conditions of soil. There are many influencing factors, such as soil density, soil structure, clay content, saturation, gradation, water permeability, initial stress state and dynamic load characteristics. Generally speaking, the mechanism of earthquake liquefaction can be expressed as follows : The shear wave propagates upward from the underlying rock and soil layer during the earthquake, and causes alternating stress in the soil, resulting in vibration pore water pressure, which is the main reason for the liquefaction of saturated sand. Under the action of alternating stress, new stress will be generated at the contact point of soil particles. When this stress reaches a certain value, the original connection and structural state between soil particles will be destroyed, and the sand particles will be separated from each other. At this time, the stress originally transmitted by sand particles through the contact point will be transmitted to the water in the pores to bear, resulting in the increase of pore water pressure. With the increase of the number of stress cycles, the pore water pressure increases due to the gradual accumulation. On the one hand, pore water tries to discharge upward under a certain vibration pore water pressure; On the other hand, soil particles try to sink down under the action of self weight, so that at the moment of structural damage or within a certain period of time, the downward sinking of soil particles is hindered by the upward discharge of pore water, so that the soil particles are partially or completely suspended, and the shear strength disappears locally or completely, which means that the soil has varying degrees of deformation or complete liquefaction. In other words, the seismic liquefaction of sand is related to the ground motion excitation and the properties of soil. The ground motion excitation here mainly refers to the peak acceleration and the duration of earthquake. For the earthquake liquefaction phenomenon in the region far away from the epicenter mentioned by the authors, being far away from the epicenter does not mean that liquefaction will not occur, and the peak acceleration in the region far away from the epicenter may not be low. Studies have found that liquefaction occurs when the earthquake intensity is as small as moment magnitude 4.5 (Green and Bommer, 2020). The existing mechanism of seismic liquefaction mentioned by the authors is based on undrained conditions.

In fact, drainage and undrained conditions are not unnecessary conditions for seismic liquefaction of sand. Undrained conditions are only to explain that pore water pressure offsets gravity. Even under drainage conditions, pore water pressure may dissipate slowly, but sand liquefaction is slow. Soil structure is also an important factor to control the liquefaction of sand. After the structure is damaged by earthquake vibration, the vibration pore water pressure gradually increases with the increase of the number of stress cycles, and its size is finally enough to make the saturated sand lose its shear capacity locally or completely. That is, the seismic liquefaction of sand is affected by the compactness of soil. Even if there is a new mechanism of sand seismic liquefaction, it should be based on particle mechanics, such as the so-called shear jamming mechanism. The innovation of this study is limited.

In general, according to the current status of this study, the reviewer regrets that, it is not recommended to be published in this professional journal.

We agree with many of the reviewer's comments describing soil liquefaction and the processes associated with it. Yet, the undrained perspective, which has been perceived as an axiom in the liquefaction research community, has been recently questioned (e.g., Goren et al 2010,2011; Madabhushi & Haigh 2012; Lakeland 2014, Odamidis & Madabhushi 2016,2018). Here, we propose the non-dimensional Deborah (De) number framework to estimate the soil drainage conditions and expected response, based on basic physics. This allows questioning the validity of the undrained assumption in specific setups, geometries, and boundary conditions. We further show that pressurization can be achieved even under well-drained conditions when the fluid flow itself produces the pressure gradients and pore pressure values needed to liquefy the soil. This should be distinguished from liquefaction initiation under undrained conditions that precede a post-seismic dissipation of pore pressure.

The current work addresses the puzzling nature of liquefaction beyond the earthquake near-field, aiming to explain and predict the many observed liquefaction events which occur under a very low seismic energy input (Wang, 2007), much below the minimum input required to liquefy soils in undrained lab tests (Green & Mitchell 2004). We show that drained liquefaction is not limited by the undrained seismic energy threshold and thus can be invoked to explain the occurrence of liquefaction under very low seismic energy input (with values typical to the earthquake's intermediate and far fields).

We think that the community will benefit from studies like this, that promote rethinking of basic assumptions like the undrained approach. Especially, since as cited above, there is already a non-negligible amount of research pointing in that direction. Our analytical approach is laid out and supported by numerical simulations and experiments. It should be noted, that we do not aim to replace the undrained mechanism, we simply demonstrate that liquefaction can be triggered under a full spectrum of drainage conditions, and that drained liquefaction can be the missing piece in liquefaction related puzzles (e.g., far-field liquefaction).

Reviewer #2 (Remarks to the Author):

The manuscript titled “Drainage explains soil liquefaction beyond the earthquake near-field” examines the phenomenon of earthquake-induced liquefaction, focusing on the importance of co-seismic drainage, with implications on the conventional view of liquefaction triggering. Methodologically, it combines analytical with numerical and experimental work. It is a very welcome addition to the growing body of work that challenges the traditional view of liquefaction as an undrained phenomenon. The results reported are noteworthy and I expect them to be of significance to the field. The work presented supports the conclusions and claims. I would recommend the manuscript for publication subject to the following points being addressed:

1. Line 109: the description of an upwards moving “compaction” front – referred to in literature also as a “solidification” front – which separates the layer into two parts, a compacted, or solidified, below the front and an uncompacted, or unsolidified, agrees with literature (also with references 28, 33, 34), though it is often referring to post-liquefaction reconsolidation, following an undrained assumption for excess pore water pressure generation.

We agree with this comment. To clarify this point, we added a paragraph at the end of section 2:

“This compaction front is in many aspects similar to previously described post-liquefaction “solidification fronts”, which are preceded by an undrained liquefaction initiation and result in consolidation (Florin & Ivanov, 1961; Wang et al. 2013). In the drained end-member described here, the compaction front forms and migrates co-seismically, and its very migration causes liquefaction above it, as demonstrated next.”

Following your compaction front analysis, how can the initial, downwards propagation of a front separating liquefied from non-liquefied soil be explained (e.g. [33])? Is this an aspect of response that can be captured with your analytical approach? This downwards propagation of a “liquefaction” front precedes the reconsolidation process, where an upwards moving “solidification” (or here “compaction”) front is observed – this reconsolidation process seems to be explained well with your compaction front analysis and indeed experimental observation shows that reconsolidation typically starts during the earthquake and not necessarily after its end, agreeing with your view of liquefaction as a “drained” phenomenon.

At this stage, our theory cannot predict analytically the downward progressing liquefaction front. However, following this suggestion, we looked for and identified the liquefaction front in our liquefied simulations. A sample simulation exhibiting the very rapid liquefaction down-going front is now shown in Fig 6, inset. Our

limited numerical resolution prevents us from studying it thoroughly. We do see that this liquefaction front controls the liquefaction onset, since the upwards compaction front initiates from the deepest point that the liquefaction front reaches. Similar to our simulations, in the experiments presented in Wang et al. (2013) and Adamidis & Madabhushi (2016) the liquefaction front is very fast and almost undetectable. Following this suggestion of the referee, we added a discussion related to the existence of the downward liquefaction front as part of sec 4.1 which discussed the initiation timescale of liquefaction:

“The timescale associated with pore pressure rise to lithostatic values in the simulations and experiments is short, and likely related to a rapid downward propagating liquefaction front. Such a behavior, viewed as “unloading” fronts (Florin & Ivanov, 1961), was previously identified in experiments (Florin & Ivanov, 1961; Sharp & Dobry, 2002; Gonzalez et al. 2005) (see Methods section 10.4 and section A in the supplementary material). We observed that the deepest location to which the down-going liquefaction front reaches correlates with the imposed shaking frequency (Appendix C), and that the liquefaction front reaches this deepest location in less than two shear cycles, consistent with previous experiments conducted under drained conditions (Wang et al., 2013; Adamidis & Madabhushi, 2016).”

In the revised version, in addition to demonstrating the presence of the downward liquefaction front in Fig. 6 (inset), we also analyze the depth to which the liquefaction front progresses in a newly added Appendix C.

Furthermore, a new Supplementary Material section is introduced, which shows the grains vertical velocity over depth and time for all of our simulations. Some of these figures allow identification of the down-going liquefaction front, despite its high speed.

2. Some further explanation on when and where the compaction front is formed when applying your analytical (or theoretical) approach would be welcome. In some cases (e.g. Fig2c, Experiment e2) only the top part of a deposit liquefies. Is it possible to apply your analytical approach to predict the greatest depth where soil is liquefied, and when this happens, or is this taken as a given and the compaction front analysis starts from there?

The initiation depth of the front is an important topic, and although it was not the focus of this study, following this comment, we added appendix C which shows that the initiation depth in simulations is proportional to the shaking frequency. In addition, as explained in our answer to the previous comment, we report a finding that the compaction front initiation depth is the deepest point that the liquefaction front reaches. This is expressed in line 516:

Our results indicate that t_i may be controlled by the descending "unloading" liquefaction front that precedes the upward compaction front. Liquefaction initiation is only complete once this unloading front reaches the compaction front initiation depth, as seen in Fig. 6 and in section A in the supplementary materials.

A possible implication for the observed frequency dependence of the initiation depth for the ascending compaction front, is that under some shaking frequencies, thinner layers will not liquefy because their thickness is smaller than the compaction front initiation height, for this frequency. This topic however merits further study, beyond the scope of this study.

3. Line 112: The assumption of stationary particles below the compaction front is essentially the assumption of Florin and Ivanov [33] but experimental data from centrifuge testing (e.g. [28], [29]) has shown that the pore water gradient below the front is not actually hydrostatic, so that further consolidation does take place below the "compaction" front, meaning that compaction and fluid flow would still take place in region 1 of Figure 1. In any case, even if the pore water gradient were hydrostatic, some compaction is inevitable as the effective stress below the compaction front gradually increases with the upwards propagation of the front. Though it is understood that some simplification is necessary, a comment to clarify to the reader the potential limitations of the assumption made would be welcome.

We agree that some compaction does occur below the front. Following this comment, we rephrased the relevant sentence using the adverbs "nearly" and "approximately" when referring to the grain layer below the (in section 2 and section 4.1). Lines 127-132 now reads:

"The propagating compaction front separates two regions within the layer (Ben-Zeev et al. 2020) (Fig. 1a-b): a lower region, in which grains have compacted to porosity $\phi_c < \phi_0$ and are approximately stationary in the vertical direction. In this lower region, the pore pressure gradient is nearly hydrostatic, although the pore pressure itself is elevated to the value of the pressure at the front (Fig. 1c)."

It is worth noting that for the physics described in the current study, stagnant grains below the front are a good approximation, as reflected by the agreement of the numerical and experimental results with the analytic predictions in figure 2a. This approximation is also validated by the newly added supplementary material section (grain velocity maps).

4. Figure 3b: is the equation plotted with a black line eq3 or eq4?

Eq. 4. Thank you for the catching the typo.

5. References [15] and [16] are cited regarding the lower bound of energy density required to cause liquefaction. Cases below 30 J m^{-3} are given the term “unexplained”. Based on this assessment, 61% of events examined in figure 4 are unexplained – an alarmingly high number. However, in practice the most common methodology to assess liquefaction triggering is the stress based simplified approach, introduced by Seed & Idriss (1967). See Boulanger & Idriss (2016) for a recent iteration of the method. This method gives a factor of safety for liquefaction occurrence. Would 61% of the events you examined remain “unexplained” if the simplified method was used? It would be very useful for the reader to know the number of “unexplained” events based on the methodology that is typically used in practice.

Seed, H. B. & Idriss, L. M. (1967). Analysis of Soil Liquefaction: Niigata Earthquake. *Journal of the Soil Mechanics and Foundations Division* 93, No. SM3, 83–108.

Boulanger, R. W. & Idriss, I. M. (2016). CPT-Based Liquefaction Triggering Procedure. *Journal of Geotechnical and Geoenvironmental Engineering* 142, No. 2, 04015065

The factor of safety in the stress-based simplified procedure is structured as the ratio between the CSR (Cyclic Stress Ratio), which represents the seismic loading on the soil in terms of acceleration/stress, and the CRR (Cyclic Resistance Ratio) which represents the soil resistance. The CRR is evaluated empirically by plotting the CSR during liquefaction/non-liquefaction vs. in situ soil resistance index (SPT-Standard Penetration Test / CPT-Cone Penetration Test/ other) from case history database, and drawing the most probable line between liquefaction to non-liquefaction. Almost by definition, events that were part of the original database from which the CRR line was constructed, are “explained” by the simplified procedure.

In that sense, it is of interest to look for new events that mismatch the original model and can be called “unexplained”. Interestingly, one of the motivations for updating the “stress based simplified approach” (Seed & Idriss 1967), mentioned by the reviewer (Boulanger et al. 2016), is the accumulation of new data over the course of time elapsed between the introduction of the approach and present time. Such periodic updating of predictions is an inherent feature of data-driven or semi-data-driven approaches like the “stress based simplified approach”. In contrast, a physics-based approach (as suggested in this study for example), provides an independent prediction, that does not need change with accumulation of data, pointing to the importance of physically-based studies.

The current study aims to provide a physical model to fill an existing gap of knowledge, and not to perform a rigorous database study or databases comparisons. The observations we use are from Wang (2007), who demonstrated that many events fall below the expected energetic threshold for undrained liquefaction, and by this identified a gap in knowledge. Although not popular as the stress-based approach, the energy approach is not new, and pursuing it as an alternative was recommended by many experts’ workshops and reviews (e.g., Youd et al. 2001, National Academies of Sciences, 2016). The energetic threshold marked

by Wang (2007, after Green & Mitchel, 2004) arises from undrained experiments, independently from the field data.

The fact that many real liquefaction events mismatch what is expected from undrained prediction (by orders of magnitude) is an excellent demonstration of the failure of the undrained approach to explain all soil liquefaction occurrences, making it a compelling observation for researchers pursuing an alternative mechanism.

That being said, we do acknowledge that the use of 'unexplained' might be misleading, and following this comment, we change the wording in lines 15&302 to 'puzzling'.

6. Line 253-254: Some clarification is needed in this phrase to avoid confusion. You state that no further significant compaction is accumulated after the end of the earthquake. You should clarify if you are referring to the whole liquefiable deposit and if so, how you explain significant post-earthquake settlements observed during post-earthquake reconsolidation e.g. [33], [28], [29].

We agree with the reviewer that this important point needs clarification.

Within the parameter range of our tests, our drained simulations show (to be published as part of a work in progress about repeated liquefaction) that the amount of post-seismic compaction is negligible compared to the co-seismic compaction/settlement. A similar observation regarding the settlement rate was already made by (Madabhushi & Haigh, 2012; Adamidis & Madabhushi, 2016) based on centrifuge tests.

The residual compaction and pressure diffusion after the cessation of seismicity is not a focus of the current study, but we can propose some scenarios where significant soil settlements in the post-seismic phase or at least long duration post-seismic settlement can occur:

The first scenario is due to a low permeability or a very thick liquified layer- After the earthquake stops, the rate in which the excess pore pressure dissipates and fluid flows out depends mainly on the permeability of the soil. Hence, long pressure dissipation is expected where the permeability is low, or the liquified layer is very deep. Both contributing to a high De number and would tend to push the event toward the undrained limit. This high De is outside the regime of our experiments and simulations. Furthermore, in a very thick layer, it is possible that minor local change in porosity, while the residual pore pressure is dissipating, will be witnessed as larger soil settlement of the total layer (by integrating small changes over the layer and accumulating a total volumetric strain).

Another possibility for witnessing a significant post-seismic settlement, is where there is a low permeability seam (which is not the case in [28,29] but is common in the field, as in [Holzer et al. 1989]). This again poses an undrained scenario, outside the scope of the present study, Even if the seam does not yield, a slow fluid flow across it will act as a pressure source for the layer above it, that can be sustained over longer durations (similar to fluidization experiment, or coastal quicksand driven by up-flow). In this scenario, the reduction of fluid volume accumulated below the seam could contribute to observations of post-seismic settlement.

Following this comment, we modified the text in section 4.2 as follows:

“An intrinsic timescale for drained liquefaction is the event termination time, t_e , which depends on the front velocity. However, in natural events, the earthquake shaking duration, t_{EQ} , is finite, and after time t_{EQ} the seismic energy input drops to zero. The post-seismic evacuation of the excess pore pressure occurs at a rate that depends mainly on the permeability and in our simulations and experiments is accompanied by negligible residual compaction.”

7. Line 357: Typically in liquefaction experiments, laminar boxes are used, where each horizontal lamination can move relative to the others. This would be closer to the periodic boundary used for the numerical simulation. Given the practically rigid box used and the small dimensions compared to similar experiments in literature (as already cited in this manuscript), a comment on potential boundary effects is necessary.

The primary purpose of the experiments was to examine the grain dynamics and fluid pressurization visually. To achieve that, we chose to use a completely transparent box to allow filming of at least one face of the box. We succeeded in achieving this goal with the current experimental setup, as we were able to validate visually the formation of the compaction front and its velocity relation to the imposed energy density (and rate of energy density input). We are aware of the limitations of this experimental setup, but it allowed us to validate the trends that emerged from theory and simulations in a way that centrifuge tests or laminar box setup would not allow. Even more so, the observation that despite their limitations (small size, boundary effects, rigid), the experiments showed the same dynamics as the simulations (the formation of the compaction front and its velocity relation with the imposed seismic energy density and rate of seismic energy input, see figs.2a-d & 3a-b,d), and also agreed perfectly with the theoretical prediction, is a strong support for both the robustness of the compaction front dynamics theory, and the suitability of the current experimental setup for studying it.

As there are limitations to any experimental choice, the experimental setup here did serve its designated purpose.

Following this comment, we added the following text to briefly express these views (section 4.1):

“...Both robust correlations emerged despite inherent differences in the boundary conditions and geometrical setup between the simulations and experiments, suggesting that, within the framework of drained liquefaction triggering, the seismic energy density, and possibly a new measure, the seismic energy density rate, can be considered as controlling parameters on the magnitude and duration of liquefaction events.”

We also note that to minimize the wall’s effect we limit the grain velocity measurements to a region close to the box center. Thank you for pointing the absence of this explanation from the manuscript. We added the following text to section 10.3:

“The instantaneous grain velocity field is measured in every frame by using PIVlab (Thielicke, & Stamhuis 2014; Thielicke & Sonntag, 2021), which relies on sub-frame correlation between timely-adjacent frames. The vertical velocity is then averaged over sub-layers, yielding the vertical velocity of grains as a function of depth and time (see section 10.4). To minimize boundary effects from the box's walls, the averaging is done only close to the box's center (approximately in the middle 2/4 of the box's total width).”

Moreover, it is implied that the input motion was harmonic but it would be better if this was clearly stated in the text.

Thank you. The revised text clarifies this point in section 10.3:

“The box is attached to a horizontal shaker (Tira S51120) that is fed with harmonic signal from a signal generator (Agilent 33220A) through an amplifier (BAA500).”

Reviewer #3 (Remarks to the Author):

This is an interesting paper that aims to explain the soil liquefaction in the far-field where the earthquake shaking intensity is small. It is true that such events have been observed in previous earthquake events and reported in case histories. Several theories exist for that such as soil amplification in some regions of the far-field that cause liquefaction while attenuation of accelerations in other regions may not lead to full liquefaction.

The following comments are offered for the authors to consider.

1. It may be better to distinguish between ‘liquefaction’ and ‘fluidisation’ throughout the manuscript. I would use ‘liquefaction’ for excess pore pressures generated due to seismic shaking and ‘fluidisation’ for loss of shear strength of soil due to pore fluid migration (from lower layers towards the soil surface) and can happen into regions where shaking intensity is not high. Authors may want to clarify that when they said ‘drained liquefaction’ if they mean ‘fluidisation’.

The term “liquefaction” or “seismically induced liquefaction” is used throughout the manuscript to describe the loss of shear strength of the soil due to seismic shaking-induced pressurization. One of our major findings is that this pressurization can be achieved via drained liquefaction, which adds to the familiar undrained end member. Distinguishing the high De liquefaction (undrained) from the low De (drained) liquefaction, by calling each a different name (the 1st “liquefaction” and the 2nd “fluidization”) will artificially differentiate between end-member behaviors that are in fact a part of a continuum and present similar field manifestations (ground settlement, wave attenuation, rigidity loss, high fluid pressure). Such an artificial division may only decrease the ability to understand and analyze liquefaction and pressurization processes as part of a continuum, as revealed by our new work.

Furthermore, the term ‘fluidization’, which usually refers to grain suspension via an externally imposed fluid flow, could be misleading in this context, since there is no imposed flow through the boundary (e.g., fluid injection at the bottom), rather the flow is the outcome of in-situ compaction-pressurization feedback across the compaction front.

To clarify the terminology confusion, we included a new appendix (A) that discusses terminology issues (also relevant to the following comment).

2. In the same vein, it may be better to distinguish clearly between the ‘drained’ and ‘undrained’ terms. In liquefaction research (see Refs 22, 24 and 29) the term ‘partially drained’ was introduced to demarcate this. In this paper the authors used the Debroah number (De) to indicate partially drained events.

The current study does not refer to “drained” and “undrained” in the context of pore pressure value, but rather by comparing the time scale of pore pressure diffusion and the time scale of the forcing inducing pore space deformation.

The perspective we attempt to bring forward as part of this work is that the non-dimensional De number can be used to adequately define the drainage conditions. The De number and the pore fluid equation (eq. 1) describe two end-members. When $De \gg 1$ the layer response is undrained, fluid cannot dissipate in the timescale of granular deformation and the diffusion term is negligible. The other end-member, where $De \ll 1$, fluid can easily flow within the timescale of pore space change, and the first term of eq. 2 is negligible, is referred to as drained.

We believe that previous uses of “partially drained” to describe scenarios of $De \ll 1$ reflect the degree to which the undrained hypothesis has become an axiom in the liquefaction research community. A prefix “partially” was felt needed to explain observations of fluid flow during liquefaction. We suggest the contrary: Two theoretical end members should be considered, “drained” and “undrained”. Once a clear definition of them is agreed, it is easier to discuss the in-between cases

(which are probably common in nature and characterized by $De \sim 1$), and they should be referred to as "partially drained" or "partially undrained".

To extend our point, we note that Darcy's law describes fluid flow (drainage) in relation to non-hydrostatic pressure gradients. I.e., one cannot categorically exclude excess pore pressure in drained conditions.

Following this comment, we added a discussion of terminology related issues in new appendix (A).

3. The authors may want to clarify what are the time scales for the liquefaction and fluidisation events. One would expect liquefaction to happen (for loose sands) in the first few cycles of earthquake shaking. Fluidisation can happen and continue into the post seismic period.

Please refer to our reply to your question 1, and reply 6 to reviewer #2. We modified Appendix B - Liquefaction initiation timescale to discuss also the time scale of the downward liquefaction front.

Generally, the time scale of liquefaction initiation is longer under undrained conditions ($De \gg 1$) and shorter under drained conditions ($De \ll 1$). This was previously shown (independent from this study) by *Adamidis & Anastasopoulos (2022)*.

Within the parameter range of our tests, our drained simulations show (to be published as part of a work in progress about repeated liquefaction) that the amount of post-seismic compaction is negligible compared to the co-seismic compaction/settlement. Consequently, the timescale of pore pressure dissipation is also short (given the characteristic permeability). Based on centrifuge tests, a similar observation regarding the post-seismic settlement rate was previously made by (Madabhushi & Haigh, 2012; Adamidis & Madabhushi, 2016).

4. The concept of 'liquefaction front' moving from soil surface towards the bedrock as the shaking starts and the 'densification front' moving from the bedrock towards the soil surface have been previously identified by many researchers (e.g. ref 33). There is a large body of dynamic centrifuge test data to show these fronts experimentally occurring in layers of ~ 10 m deep. The authors may want to include some of these references into the manuscript.

We agree with this comment. Please see our reply to reviewer #2 1st question and the related text changes we introduced following these comments.

5. There is considerable body of research on liquefaction that establishes cyclic shear strain amplitude threshold i.e. earthquakes that cause shear strains above this threshold cause excess pore pressure generation while those below won't. I feel this is more physical as shear strains are physically linked to the movement of individual soil grains and the length parameter.

Incidentally I agree with the authors argument about length scale in Sec. 3.1. This almost corresponds to Tatsuoka's observations of formation of shear bands and their thickness in sands (~20 particle diameters). However, I do not see why the energy argument is used in this paper. Measures like Arias intensity that depend on the PGV are global measures, where as liquefaction (in my view) depends on the relative displacement between particles. I am not talking about fluidisation here (as distinguished in point 1 above). The authors may want to argue why PGV is a good measure for liquefaction or otherwise.

We agree that liquefaction needs volumetric strain to occur. The relative motion of grains beyond elastic limits is the response of the soil to external excitation, which in this paper, we found to be related to the seismic power and the seismic energy density (scales with PGV^2) This is in agreement with Goren et al. (2010,2011) who showed that unlike undrained conditions, where the volumetric strain control liquefaction, the drained response depends on the strain rate (which scales with PGV).

Furthermore, an energy-based approach for the evaluation of liquefaction potential is not new, and pursuing it as an alternative to stress-based and strain-based approaches was recommended by experts' workshops and reviews (e.g., Youd et al. 2001, NAS 2016).

Specifically, the energy-based measure Arias Intensity was previously shown to correlate with liquefaction potential (Kayen & Mitchell, 1997). We believe that it is likely a good predictor for the undrained end-member. We found that for the drained end-member studied here, the best controlling parameter is similar to Arias Intensity per cycle when considering a harmonic signal (The seismic power). This view is now communicated in the revised version (section 4.1 line 242):

“A leading energy-based approach for evaluating soil liquefaction potential uses the earthquake's Arias intensity (Kayen & Mitchell, 1997). While the Arias intensity is a cumulative measure that accounts for the amplitude and frequency content throughout the duration of the earthquake, the rate of seismic energy density input (e/T), which we consider here, can be interpreted as a quasi-instantaneous Arias intensity or an average power of ground shaking over one shear cycle. The excellent performance of the rate of seismic energy density input in explaining the amount of compaction (Fig. 3b) and the front velocity (Fig. 3d inset), and in defining the clearest threshold between liquefied and non-liquefied simulations (Fig. 3b) suggests that a dynamic process as drained liquefaction depends on the momentary power rather than on the cumulative power. This is likely in contrast to undrained liquefaction, which is a cumulative process in nature (the volumetric strain required to initiate liquefaction is accumulated over many shear cycles (de Alba et al., 1976), hence it might depend on a cumulative energy measure like Arias intensity.”

6. In Sec. 4.1 the authors argue that the time it takes to reach 'full drained liquefaction' is exceedingly small i.e. $10^{-3} \sim 10^{-8}$ seconds. I presume that the authors mean that this is different from 'seismic liquefaction' and are saying fluidization can occur much more quickly.

Firstly, for seismic liquefaction, the excess pore pressure generation can take anything from 1 to 2 cycles to many cycles (~8 to 10 cycles) depending on the relative density of the sand. Loose sands liquefy more easily than dense sands. Secondly, the time for fluidisation surely depends on the permeability, porosity of the soil strata and pressure gradient applied which will yield much longer time scales, as pore fluid has to migrate from one region to the other?

Following this comment, we edited the mentioned paragraph as follows:

“The timescale associated with pore pressure rise to lithostatic values in the simulations and experiments is short, and likely related to a rapid downward propagating liquefaction front. Such a behavior, viewed as “unloading” fronts (Florin & Ivanov, 1961), was previously identified in experiments (Florin & Ivanov, 1961; Sharp & Dobry, 2002; Gonzalez et al. 2005) (see Methods section 10.4 and section A in the supplementary material). We observed that the deepest location to which the down-going liquefaction front reaches correlates with the imposed shaking frequency (Appendix C), and that the liquefaction front reaches this deepest location in less than two shear cycles, consistent with previous experiments conducted under drained conditions (Wang et al., 2013; Adamidis & Madabhushi, 2016). Another timescale operating in the system is the time required for an isolated, fully immersed, grain (equation 5, Appendix B) to reach its terminal downward velocity. However, since this timescale is exceedingly small, $10^{-3} - 10^{-8}$ seconds, the acceleration of a single grain is not a rate-limiting process for triggering drained liquefaction. Recent cyclic triaxial experiments (Adamidis & Anastasopoulis, 2022) found that the number of cycles required to initiate liquefaction under drained conditions is smaller than under undrained conditions, supporting the hypothesis that a pressurization time of the order of a few cycles could be indicative of drained liquefaction initiation. Such a consideration might apply to a recent ground motion analysis showing that, in some cases, the time for liquefaction triggering is as short as ~ 1.7 seconds from the onset of recorded shaking (Greenfield, 2017; Ozener et al., 2020).”

We stress that in our view, the term “fluidization” does not adequately describe our results, since no pressure gradient is imposed externally. Although we did not explore a wide range of permeabilities in the current study (the simulations and experiments had different permeabilities), a previously published study (Ben-Zeev et al., 2020) found similar dynamics over a wide range of permeabilities.

7. I have several objections on the experimental program reported in Sec. 10. Also not sure why the authors do not use data from dynamic centrifuge tests on liquefaction.

The main purpose of the experiments was to examine visually the grain dynamics and fluid pressurization through the experiments. To achieve that, we chose to use a completely transparent box to allow filming at least one face of the box. This goal was achieved as we were able to validate visually the formation of the compaction front and its velocity in relation with the imposed power and energy. Despite the limitations of this experimental setup (small size, boundary effects, rigid), it allowed us to validate the trends that emerged from theory and simulations in a way that centrifuge tests or laminar box setup would not allow. The experimental results agree very well with simulations (the formation of the compaction front and its velocity relation with the imposed energy and power, see figs.2a-d & 3c). The fact that the same dynamics emerged despite the difference in setup between simulations and experiments, and the fact that both experiments and simulations agree with theory, shows that the compaction front dynamics are robust, and the experimental setup, in spite of its limitations, suffices to capture it.

To convey this message, we added in the discussion (section 4.1) the following sentence:

“Both robust correlations emerged despite inherent differences in the boundary conditions and geometrical setup between the simulations and experiments, suggesting that, within the framework of drained liquefaction triggering, the seismic energy density, and possibly a new measure, the seismic energy density rate, can be considered as controlling parameters on the magnitude and duration of liquefaction events.”

a) The experiments are in a very small-scale model container 120 mm (cube). The boundary effects will be significant in such a small model container.

We calculate velocities in the middle part of the box’s face. We note that despite the box’s small dimensions, the same dynamics emerged as in simulations, and both experiments and simulations show remarkable agreement with our liquefaction theory, suggesting the experimental setup is indeed sufficient to produce the liquefaction process.

Following this comment, we added information in section 10.3:

“The instantaneous grain velocity field is measured in every frame by using PIVlab (Thielicke, & Stamhuis 2014; Thielicke & Sonntag, 2021), which relies on sub-frame correlation between timely-adjacent frames. The vertical velocity is then averaged over sub-layers, yielding the vertical velocity of grains as a function of depth and time (see section 10.4). To minimize boundary effects from the box's walls, the averaging is done only close to the box's center (approximately in the middle 2/4 of the box's total width).”

b) What is the relative density of the soil specimens? This should be included in Table 2.

Porosity is not convenient for this as RD is measure of the soil state with respect to the loosest and densest possible states.

We are aware of the costume to report the relative density of the soil specimens in the geotechnical community and consequently we would have like to follow the reviewer recommendation. However, this very study (see fig. 3a) has revealed to us that the densest state of a soil specimen is a function of the shaking energy and the rate of seismic energy input (see fig. 3b), and therefore, the relative density does not have a unique value. On the other hand, porosity is a uniquely defined physical quantity, that appears in the theory we developed (from basic physics).

c) The frequency of shaking is very high (10 Hz, most earthquakes are in the 1 to 5 Hz range)

10 Hz is indeed in the higher range of expected earthquake frequencies (though certainly not the maximum frequency, see for example (Souriau) 2006 and (Tosi et al. (2012)). In order to experimentally explore seismic energy density values in a range similar in magnitude to the seismic energy density in simulations, and to ensure the low seismic energy density range representative to the far field, we had to choose a high frequency for the experiments to overcome limitations on the maximum allowed displacement of our shaker. The agreement of our results with theory reassured that the high frequency did not effect the physics considerably.

Yet, following this comment we did indeed probe more deeply the effect of frequency on our results. We found that one effect of frequency is on the initiation depth of the compaction front. We thus added a new Appendix C showing the initiation depth as function of frequency.

d) The amplitude of shaking is very small 0.8 mm is the largest amplitude used. How does this compare to the mean particle size (D20 or D50) for this sand?

the mean grain diameter in the experiments (reported in Table 1) is 0.02 cm and the minimum shaking amplitude is 0.024 cm (reported in Table 3), i.e., the shaking amplitude is at least of the order of the mean grain size.

Following this comment, we expanded the method section (10.3) to address this point:

“Horizontal shaking is applied for 30 seconds with displacement amplitude of at least one mean grain diameter.”

e) Referring to Fig.2a, the excess pore pressures do not show any suction cycles, cyclic oscillations (to indicate the crossing of phase transformation line (following Ishihara) nor reach full liquefaction of 1. The authors may want to explain why.

The pressure signal is smoothed in order to highlight the most important trend (and it is a very noisy signal). Furthermore, we see in others non-jacketed tests (e.g., centrifuge) that suction cycles are not observed (for example in Wang et al. (2013) and Adamidis & Madabhushi (2016))

The fact that the pressure ratio does not reach 1 can be explained by a number of reasons. First, the errorbar (pink shaded area in fig.2a) is calculated based on a static calibration procedure during the pre-shaking stage. It is possible that the errorbars should be larger during the dynamic stage of shaking. I.e., the error shown is the minimal one.

Alternatively, the pressure in fig. 2a is normalized by the initial static solid stress which is evaluated analytically from porosity, grain density and depth below the surface. The pressure sensors remain in place during the experiment (they are attached to the box's face), but since during drained liquefaction the grains settle, the overburden at the sensor depth is expected to decrease (relative to its initial value). This will bring the curve closer to 1 (dividing the pressure by a smaller number). Another possible effect is that the sidewalls (which are absent in simulations) carry some constant amount of the soil load, instead of the fluid.

Together with the above suggestions, the fact that the pressure value reaches a plateau before dropping back to zero might indicate that the pore pressure ratio reached the value of 1 during the liquefied state. A stable pressure value is easier to maintain at a complete liquefaction. Despite these suggestions, to remain conservative, we prefer to keep the current form of the pressure curve.

f) The time for generation of excess pore pressure in Fig. 2a is much more slower than in the simulations and magnitudes are smaller.

The time axis in figure 2a is normalized by the duration of the liquefaction event, which is not identical in the simulation and the experiment presented. Please see the figure below, showing that the number of cycles until complete liquefaction is $N=2$ in the simulation and $N=7$ in the experiment. The time until complete liquefaction is 0.34 and 0.65 seconds, respectively. This difference is attributed to the delay of the shaker before reaching the designated amplitude (due to inertia and friction), unlike in simulations where the imposed shaking BC is immediately applied.

g) In fig. 2d, the shear strains are biased towards the positive side? Why?

This is due to a complication arising when defining shear strain in an unconfined layer. If the layer is confined between two sheared plates, then it is easy to define the shear strain in the layer as the relative motion between the two plates (where the shear is imposed) divided by the layer's thickness. Here, with the free surface at the top, we define the shear strain in each sub-layer as the difference of the horizontal velocity of grains integrated over time, divided by the vertical thickness of that sub-layer. The data presented in figure 2d is from the middle of the simulation box. The strain axis being not symmetric around zero is a remnant of the initial effect where the grains were moving in the positive direction before liquefying. Also, the imposed motion is of $(1-\cos)$ which is not centered around zero.

h) In Fig.2d, are the experimental results and simulations at the same frequency? The recoupling and increase in oscillations in the experiment are not clear.

We assume you refer to fig 2c. The simulation is of ~ 6.7 Hz and experiment of 10 Hz. The time axis is normalized by the total duration of the liquefaction event, which is different between the presented simulation and experiment. Indeed, the change in amplitude of motion of the experiment grains is less dramatic than in the simulations. This is likely because in the experiments, the sidewalls impose

horizontal motion even to the liquefied layer, unlike simulations where the motion is imposed only at the bottom. Despite this behavior, there is a clear trend in the experiments relating to regaining 60 % of the imposed shear amplitude. Following this comment, to achieve clarity, we added a magnification inset around the front in the experiment panel of fig.2c.

References

- Goren, L., Aharonov, E., Sparks, D. W. & Toussaint, R. Pore pressure evolution in deforming granular material: A general formulation and the infinitely stiff approximation. *Journal of Geophysical Research: Solid Earth* 115,1–19 (2010).
- Goren, L., Aharonov, E., Sparks, D. W. & Toussaint, R. The Mechanical Coupling of Fluid-Filled Granular Material Under Shear. *Pure and Applied Geophysics* 168, 2289–2323 (2011).
- Madabhushi, G. S. P. & Haigh, S. K. How Well Do We Understand Earthquake Induced Liquefaction? 42,150-160 (2012).
- Lakeland, D. L., Rechenmacher, A. & Ghanem, R. Towards a complete model of soil liquefaction: the importance of fluid flow and grain motion. *Proceedings of the royal society* 470 (2014).
- Adamidis, O. & Madabhushi, G. S. P. Post-liquefaction reconsolidation of sand. *Proceedings of the royal society* 472 (2016).
- Adamidis, O. & Madabhushi, G. Experimental investigation of drainage during earthquake-induced Experimental investigation of drainage during earthquake-induced liquefaction. *Geotechnique* 68, 655-665 (2018).
- Wang, C. Y. Liquefaction beyond the Near Field. *Seismological Research Letters* 78, 512-517 (2007).
- Green, R. A. & Mitchell, J. K. Energy-based evaluation and remediation of lique_able soils. *Geotechnical Special Publication* 1961-1970 (2004).
- Wang, B. et al. Excess pore pressure dissipation and solidi_cation after liquefaction of saturated sand deposits. *Soil Dynamics and Earthquake Engineering* 49, 157-164 (2013).
- Florin, V. & Ivanov, P. Liquefaction of Saturated Sandy Soils. In *Proceedings of the 5th international conference on soil mechanics and foundation engineering*, 107-11 (1961).
- Sharp, M. k. & Dobry, R. sliding block analysis of lateral spreading based on centrifuge results. *International journal of physical modeling in geotechnics* 13-32 (2002).
- Gonzalez, L., Abdoun, T., Zeghal, M., Kallou, V. & Sharp, M. k. Physical modeling and visualization of soil

liquefaction under high confining stress. *Earthquake Engineering and Engineering Vibration* 4, 47-57 (2005).

Ben-Zeev, S., Aharonov, E., Toussaint, R., Parez, S. & Goren, L. Compaction front and pore fluid pressurization in horizontally shaken drained granular layers. *Physical Review Fluids* 054301, 1-25 (2020).

Youd, T. L. et al. Liquefaction resistance of soils: Summary report from the 1996 NCEER and 1998 NCEER/NSF Workshops on Evaluation of Liquefaction Resistance of Soils. *Journal of Geotechnical and Geoenvironmental Engineering* 127, 817-833 (2001).

National Academies of Sciences, Engineering & Medicine. State of the Art and Practice in the Assessment of Earthquake-Induced Soil Liquefaction and Its Consequences. Tech. Rep., Washington, DC (2016).

Thielicke, W. & Stamhuis, E. J. PIVlab - Towards User-friendly, Affordable and Accurate Digital Particle Image Velocimetry in MATLAB. *Journal of Open Research Software* 2 (2014).

Thielicke, W. & Sonntag, R. Particle Image Velocimetry for MATLAB: Accuracy and enhanced algorithms in PIVlab. *Journal of Open Research* 9 (2021).

Greenfield, M. W. Effects of long-duration ground motions on liquefaction hazards. Ph.D. thesis, University of Washington (2017).

Ozener, P. T., Greenfield, M. W., Sideras, S. S. & Kramer, S. L. Identification of time of liquefaction triggering. *Soil Dynamics and Earthquake Engineering* 128 (2020).

Souriau, A. (2006), Quantifying felt events: A joint analysis of intensities, accelerations and dominant frequencies, *J. Seismol.*, 10, 23–38, doi:10.1007/s10950-006-2843-1.

Tosi, P., Sbarra, P., & De Rubeis, V. (2012). Earthquake sound perception. *Geophysical Research Letters*, 39(24).

Kayen, R. E. & Mitchell, J. K. Assessment of liquefaction potential during earthquakes by Arias intensity. *Journal of Geotechnical and Geoenvironmental Engineering* 123, 1162-1174 (1997).

de Alba, P. A., Chan, K. C. & Seed, H. B. Sand liquefaction in large-scale simple shear tests. *Journal of Geotechnical and Geoenvironmental Engineering* 102, 909-927 (1976).

Reviewers' comments:

Reviewer #2 (Remarks to the Author):

The authors have meaningfully addressed all comments from the reviewing process. The resubmitted manuscript is improved as a result and I would support its publication.

Reviewer #3 (Remarks to the Author):

The authors have addressed many of the comments I have raised. I would still stand by my comments that it would have perhaps been better to compare the authors theory against dynamic centrifuge test data with PIV analyses, rather than their small scale experiments. Notwithstanding that, I am happy with the changes they have made.

Reviewer #4 (Remarks to the Author):

This paper deals with the important topic of liquefaction under drained conditions. Unfortunately, its contribution seems unclear and untimely. It is not clear from the manuscript what exactly is the contribution here. For example, the authors claim:

"The discovery that high permeability, well-drained soils should not be a priori assumed liquefaction resistant" That is not a "discovery" since that is well known in the literature. There have been at least one decade worth of studies on static liquefaction and drained instabilities showing that the statement above is correct.

Also, the authors rely on a coupled solid-fluid simulation that is based on spherical DEM. On the other hand, the experiments are done using a transparent tank filled with sand and water. The ability to compare between simulations and experiments here is at best qualitative. Of course, a loose packing of spheres or sand under unconfined conditions can display instabilities, especially towards the top of the column where effective stresses approach zero anyway. Any excitation that would increase excess pore pressure would tend to provoke instabilities. Again, this is not deep or new.

Unfortunately this study as is cannot be accepted for publication given its inability to clearly communicate its contribution and its timelines.

Dear reviewers,

We wish to thank you for putting the time and effort in reviewing our manuscript.

The manuscript's main objective is to solve the formerly puzzling observations of liquefaction in the earthquake's intermediate and far fields, under low seismic energy density. To achieve this objective, we harness a mechanism for liquefaction under drained conditions, which is not limited by the same seismic energy threshold as undrained liquefaction. We further show that the unique characteristics of drained liquefaction, i.e., co-seismic heterogeneous compaction controlled by the compaction front model (whose magnitude and velocity are related to the seismic excitation magnitude), can be invoked to explain the decay in the number of liquefaction observations with distance from earthquakes epicenters.

To address the main issue raised in the last round of review, the relatively small size of our experimental setup, we wrote a new appendix that compares our findings to previously published large-scale simulations and experiments. We also rewrote the entire conclusions section to better emphasize the "take home message" of the manuscript.

Sincerely,

Shahar Ben-Zeev on behalf of the co-authors

Point by point response:

Reviewer #2 (Remarks to the Author):

The authors have meaningfully addressed all comments from the reviewing process. The resubmitted manuscript is improved as a result and I would support its publication.

We are happy that our revisions were found to be satisfactory for publication, and thank Rev#2 for meaningful comments.

Reviewer #3 (Remarks to the Author):

The authors have addressed many of the comments I have raised. I would still stand by my comments that it would have perhaps been better to compare the authors theory against dynamic centrifuge test data with PIV analyses, rather than their small scale experiments. Notwithstanding that, I am happy with the changes they have made.

We are happy that our revisions were found to be satisfactory for publication, and thank Rev#3 for meaningful comments.

Regarding the comment that it is desirable to compare our results to large scale centrifuge experiments. Our physics-based theory, which is scale-less and fits well our experimental and computer modeling results, serves to validate the

applicability of our results to all scales, including large scale. Yet the reviewers' request to compare our theory and simulations also to large scale experiments is understandable. To this end we have searched previously published experimental literature, and were able to identify two manuscripts reporting large-scale centrifuge experiments (but without PIV) and simulations under enhanced gravity (simulating greater stresses and depths). These previous studies reported observables that allow us to partly compare the dynamics they observed to our theory and observations.

We added a new appendix E (called "High gravity experiments and simulations support depth independent triggering of drained liquefaction"), in which we compare different aspects of our predictions and results to the centrifuge experiment (50 g) of Adamidis & Madabhushi (2018) and to the high gravity simulations (30 g) of El Shamy & Zeghal (2007). For the latter, we show the existence of a co-seismic compaction front (triggered at prototype depth of 5.2 meters) with a front propagation velocity that fits very well our prediction for the relation between front velocity and both the imposed seismic energy density and rate of seismic energy density. Overall, these two studies support depth-independence (and consequently stress-independence) of the model proposed in our manuscript, and confirm the applicability of our model, experiments, and theory to the large-scale.

Reviewer #4 (Remarks to the Author):

1) This paper deals with the important topic of liquefaction under drained conditions. Unfortunately, its contribution seems unclear and untimely. It is not clear from the manuscript what exactly is the contribution here. For example, the authors claim: "The discovery that high permeability, well-drained soils should not be a priori assumed liquefaction resistant" That is not a "discovery" since that is well known in the literature. There have been at least one decade worth of studies on static liquefaction and drained instabilities showing that the statement above is correct.

We thank reviewer #4 for his review and comments.

We agree that the above phrasing (from the Conclusions Section) was misleading, and should not have appeared in the text. The manuscript actually is not centered around the "discovery" of drained liquefaction: Already in the introduction, we cite studies that discuss the possibility of drained liquefaction (e.g., Madabhushi & Haigh, 2012; Lakeland et al., 2014; Adamidis & Madabhushi, 2018; Adamidis et al., 2022), as well as studies that we ourselves have published starting from 2011 on drained liquefaction. The reviewer's comment made us realize that the entire conclusions section should be improved, and we rewrote it entirely, to better reflect the content of the paper and its title. The main take home message of this manuscript is not the mere existence of drained liquefaction, but its ability to explain previously puzzling far-field liquefaction events.

2) Also, the authors rely on a coupled solid-fluid simulation that is based on spherical DEM. On the other hand, the experiments are done using a transparent tank filled with sand and water. The ability to compare between simulations and experiments here is at best qualitative.

Hundreds of previously published papers quantitatively compare physical granular experiments (with and without water) to DEMs, and we fail to see the problem in that. We explicitly make a quantitative comparison between simulations and experiments in Fig 3d, showing the seismic energy density range over which the simulations and experiments were conducted (far-field energies) and their general agreement regarding the dependency between the compaction front velocity and the imposed seismic energy density. The general quantitative agreement between simulations and experiments, and also the excellent fit between these 2 scientific methodologies and our physics-based theory, demonstrate that our findings are robust, and can be identified even if the experimental setup is not identical in all minute details to the simulation's setup. In fact, the agreement between simulations and experiments performed with different materials and boundary conditions points to the universality of the process - a clear advantage and strength of the model.

3) Of course, a loose packing of spheres or sand under unconfined conditions can display instabilities, especially towards the top of the column where effective stresses approach zero anyway. Any excitation that would increase excess pore pressure would tend to provoke instabilities. Again, this is not deep or new.

This comment suggests that our observed liquefaction is due to the low stresses that characterize small-scale experiments. First, we note that we demonstrated in the manuscript the four main seismically induced liquefaction indicators in the field and lab. The liquefaction starts at the base of the grain column, ~1 m below the surface in the simulations (not so shallow). Second, we note again that our physics-based theory predicting drained liquefaction is scale-less, and thus does not require or assume small stresses. Finally, as detailed in our answer to Reviewer 3, following this concern we searched the literature and were able to identify two previously published manuscripts that conducted similar experiments (but without PIV) and simulations under enhanced gravity. These studies reported some observables that allow us to partially compare the dynamics they observed to our theory and observations. The enhanced gravity results in higher stresses and greater prototype depths.

We added a new appendix E, called "High gravity experiments and simulations support depth independent triggering of drained liquefaction", in which we compare different aspects of our predictions and results to the centrifuge experiment (50 g) of Adamidis & Madabhushi (2018) and to high gravity simulations (30 g) of El Shamy & Zeghal (2007). For the latter, we show the existence of a co-seismic compaction front (triggered at prototype depth of 5.2 meters) with a velocity that fits very well our prediction for the relation between front velocity and the imposed seismic energy density and the input rate of seismic energy density. Overall, these two studies indicate that the model proposed in our manuscript operates also at greater soil depths and at high stress levels.

4) Unfortunately this study as is cannot be accepted for publication given its inability to clearly communicate its contribution and its timelines.

The processes that lead to soil liquefaction are not yet completely understood, while liquefaction risk is evident and expected to increase (due to current and planned massive sea reclamation projects around the world). In this manuscript we attempted to provide an answer to how soil liquefaction is initiated at the far-field,

despite a very low seismic energy density excitation (Wang & Manga, 2021). We showed that if liquefaction occurs as a drained phenomenon, it possesses unique characteristics that provide a possible answer to this question. Hence, we think that this study is timely and can impact liquefaction studies as well as hazard assessment practices. In addition, following this comment, we rewrote the entire Conclusions Section, to better communicate our contributions.

References:

Kutter, B. L., Manzari, M. T. & Zeghal, M. Model Tests and Numerical Simulations of Liquefaction and Lateral Spreading - LEAP-UCD-2017 (Springer Open, 2020).

Adamidis, O. & Madabhushi, G. Experimental investigation of drainage during earthquake-induced Experimental investigation of drainage during earthquake-induced liquefaction. *Geotechnique* 68, 655–665 (2018).

El Shamy, U. & Zeghal, M. A micro-mechanical investigation of the dynamic response and liquefaction of saturated granular soils. *Soil Dynamics and Earthquake Engineering* 27, 712–729 (2007).

Adamidis, O. & Madabhushi, G. Experimental investigation of drainage during earthquake-induced Experimental investigation of drainage during earthquake-induced liquefaction. *Geotechnique* 68, 655–665 (2018).

Lakeland, D. L., Rechenmacher, A. & Ghanem, R. Towards a complete model of soil liquefaction: the importance of fluid flow and grain motion. *Proceedings of the royal society* 470 (2014).

Madabhushi, G. S. P. & Haigh, S. K. How well do we understand earthquake induced liquefaction? *Indian Geotechnical Journal* 42, 150–160 (2012).

Adamidis, O. & Anastasopoulos, I. Cyclic liquefaction resistance of sand under a constant inflow rate. *Geotechnique* (2022).

Wang, C. Y. & Manga, M. Chapter 11: Liquefaction. In *Water and Earthquakes*, chap. 11 (Springer Cham, 2021).

REVIEWERS' COMMENTS

Reviewer #3 (Remarks to the Author):

The authors have carried out comparison with previously published high gravity centrifuge test data and included in the Appendix E. I am happy with the changes they have made.

Reviewer #4 (Remarks to the Author):

The authors have significantly improved the manuscript, including its main contribution. However, there is one item that still needs to be solved. It has been made worse. The authors "fail to see what the problem is" to use classic spherical DEM to compare with experiments using sands. The effect of particle shape has been clearly demonstrated. DEM has remained qualitative until very recently. At the very least, the authors must acknowledge this explicitly in the manuscript. Current spherical simulations are qualitative and require tuning of ad-hoc parameters like damping and rotational resistance to capture the effect of particle shape. This has been documented in recent publications such as Kawamoto et al. 2018 (All you need is shape. JMPS). Once this issue is resolved by inserting the appropriate remark in the manuscript, the paper could be accepted for publication.

Point-by-point response to the reviewers' comments:

Manuscript title: "Drainage explains soil liquefaction beyond the earthquake near-field".

Reviewer #3 (Remarks to the Author):

The authors have carried out comparison with previously published high gravity centrifuge test data and included in the Appendix E. I am happy with the changes they have made.

We thank you for helping improve our manuscript.

Reviewer #4 (Remarks to the Author):

The authors have significantly improved the manuscript, including its main contribution. However, there is one item that still needs to be solved. It has been made worse. The authors "fail to see what the problem is" to use classic spherical DEM to compare with experiments using sands. The effect of particle shape has been clearly demonstrated. DEM has remained qualitative until very recently. At the very least, the authors must acknowledge this explicitly in the manuscript. Current spherical simulations are qualitative and require tuning of ad-hoc parameters like damping and rotational resistance to capture the effect of particle shape. This has been documented in recent publications such as Kawamoto et al. 2018 (All you need is shape. JMPS).

Once this issue is resolved by inserting the appropriate remark in the manuscript, the paper could be accepted for publication.

We thank you for helping improve our manuscript. We edited the following paragraph in section 2.5:

"The numerically-identified low liquefaction triggering threshold might be partially related to model parameters, including the use of perfectly smooth and spherical grains. More realistic grain shapes could potentially necessitate a higher triggering threshold⁶¹. Nevertheless, our experiments with natural sand grains showed liquefaction triggering at PGA values as low as $A_w/g \approx 0.1$ (the precise threshold was not investigated). This suggests that the intrinsic characteristics of natural grains are unlikely to alter the outcomes significantly, permitting low PGA liquefaction triggering under real-world conditions. Furthermore, after the triggering phase, once granular contacts are minimized, the dynamics of the liquefied layer are expected to be independent of grain shape, as evidenced by the congruence of front velocities across simulations, experimental results, and theory (depicted in Fig. 3d)."

Reference:

[61] Kawamoto, R., And`o, E., Viggiani, G. & Andrade, J. E. All you need is shape: Predicting shear banding in sand with ls-dem. Journal of the Mechanics and Physics of Solids 111, 375–392 (2018). URL: <https://www.sciencedirect.com/science/article/abs/pii/S0022509617306580>